# EFFECTS OF RANDOM EDGE-DROPPING ON OVER-SQUASHING IN GRAPH NEURAL NETWORKS

## ABSTRACT

Message Passing Neural Networks (MPNNs) are a class of Graph Neural Networks (GNNs) that leverage the graph topology to propagate messages across increasingly larger neighborhoods. The message-passing scheme leads to two distinct challenges: *over-smoothing* and *over-squashing*. While several algorithms, e.g. DropEdge and its variants – DropNode, DropAgg and DropGNN – have successfully addressed the over-smoothing problem, their impact on over-squashing remains largely unexplored. This represents a critical gap in the literature as failure to mitigate over-squashing would make these methods unsuitable for long-range tasks. In this work, we take the first step towards closing this gap by studying the aforementioned algorithms in the context of over-squashing. We present novel theoretical results that characterize the negative effects of DropEdge on sensitivity between distant nodes, suggesting its unsuitability for long-range tasks. Our findings are easily extended to its variants, allowing us to build a comprehensive understanding of how they affect over-squashing. We evaluate these methods using real-world datasets, demonstrating their detrimental effects. Specifically, we show that while DropEdge-variants improve test-time performance in short-range tasks, they deteriorate performance in long-range ones. Our theory explains these results as follows: random edge-dropping lowers the effective receptive field of GNNs, which although beneficial for short-range tasks, misaligns the models on long-range ones. This forces the models to overfit to short-range artefacts in the training set, resulting in poor generalization. Our conclusions highlight the need to re-evaluate various methods designed for training deep GNNs, with a renewed focus on modelling long-range interactions.

## 1 INTRODUCTION

Graph-structured data is ubiquitous – it is found in social media platforms, online retail platforms, molecular structures, transportation networks, and even computer systems. *Graph neural networks* (GNNs) (Li et al., 2016; Scarselli et al., 2009) are powerful neural models developed for modelling graph-structured data, and have found applications in several real-world scenarios (Gao et al., 2018; Monti et al., 2017; Wale & Karypis, 2006; Ying et al., 2018; You et al., 2020a;b;c; 2022; Zheng et al., 2022; Zitnik & Leskovec, 2017). A popular class of GNNs, called *message-passing neural networks* (MPNNs) (Gilmer et al., 2017), recursively process neighborhood information using message-passing layers. These layers are stacked to allow each node to aggregate information from increasingly larger neighborhoods, akin to how *convolutional neural networks* (CNNs) learn hierarchical features for images (LeCun et al., 1989). However, unlike in image-based deep learning, where *ultra-deep* CNN architectures have led to performance breakthroughs (He et al., 2016; Szegedy et al., 2015), shallow GNNs often outperform deeper models on many graph learning tasks (Zhou et al., 2021b). This is because deep GNNs suffer from unique issues like *over-smoothing* (Oono & Suzuki, 2020) and *over-squashing* (Alon & Yahav, 2021), which makes training them notoriously difficult.

Over-smoothing refers to the problem of node representations becoming *too similar* as they are recursively processed. This is undesirable since it limits the GNN from effectively utilizing the information in the input features. The problem has garnered significant attention from the research community, resulting in a suite of algorithms designed to address it (Rusch et al., 2023). Amongst

these methods are a collection of random edge-dropping algorithms, including DropEdge (Rong et al., 2020) and its variants – DropNode (Feng et al., 2020), DropAgg (Jiang et al., 2023) and DropGNN (Papp et al., 2021) – which act as *message-passing reducers*.

The other issue specific to GNNs is over-squashing. In certain graph structures, neighborhood size grows exponentially with distance from the source (Chen et al., 2018b), causing information to be lost as it passes through graph bottlenecks (Alon & Yahav, 2021). This limits MPNNs' ability to enable communication between distant nodes, which is crucial for good performance on long-range tasks. To alleviate over-squashing, several graph-rewiring techniques have been proposed, which aim to improve graph connectivity by adding edges in a strategic manner[1] (Alon & Yahav, 2021; Black et al., 2023; Deac et al., 2022; Karhadkar et al., 2023; Nguyen et al., 2023). In contrast, the DropEdge-variants only remove edges, which should, in principle, amplify over-squashing levels.

The empirical evidence in support of methods designed for training deep GNNs has been majorly collected on short-range tasks. That is, it simply suggests that *these methods prevent loss of local information, but it remains inconclusive if they facilitate capturing long-range interactions (LRIs)*. Of course, on long-range tasks, deeper GNNs are useless if they cannot capture LRIs. This is especially a concern for DropEdge-variants since evidence suggests that alleviating over-smoothing with graph rewiring could exacerbate over-squashing (Giraldo et al., 2023; Nguyen et al., 2023).

**Theoretical Contributions**. In this work, we uncover the effects of random edge-dropping algorithms on over-squashing in MPNNs. By explicitly computing the expected *sensitivity* of the node representations to the node features (Topping et al., 2022) (inversely related to over-squashing) in a linear GCN (Kipf & Welling, 2017), we show that these methods provably reduce the *effective receptive field* of the model. Precisely speaking, the rate at which sensitivity between distant nodes decays is *polynomial* w.r.t. the dropping probability. Finally, we extend the existing theoretical results on sensitivity in nonlinear MPNNs (Black et al., 2023; Di Giovanni et al., 2023; Xu et al., 2018) to the random edge-dropping setting, again showing that these algorithms exacerbate the over-squashing problem.

**Experimental Results**. We evaluate the DropEdge-variants on node classification tasks using GCN and GAT (Veličković et al., 2018) architectures. Specifically, we assess their performance on homophilic datasets – Cora (McCallum et al., 2000) and CiteSeer (Giles et al., 1998) – which represent short-range tasks, and heterophilic datasets – Chameleon, Squirrel, TwitchDE (Rozemberczki et al., 2021) – which correspond to long-range tasks. Our results indicate an increasing trend in test accuracy for homophilic datasets and a declining trend for heterophilic datasets as the dropping probability increases. Accordingly, we hypothesize that edge-dropping algorithms improve performance on short-range tasks, as has been reported earlier, by reducing the receptive field of the GNN and increasing model-dataset alignment, However, for long-range tasks, they decrease the model-dataset alignment, resulting in poor generalization.

## 2 BACKGROUND

Consider a directed graph $\mathcal{G} = (\mathcal{V}, \mathcal{E})$, with $\mathcal{V} = [N] := \{1, \ldots, N\}$ denoting the node set and $\mathcal{E} \subset \mathcal{V} \times \mathcal{V}$ the edge set; $(j \rightarrow i) \in \mathcal{E}$ if there's an edge from node $j$ to node $i$. Let $\boldsymbol{A} \in \{0, 1\}^{N \times N}$ denote its adjacency matrix, such that $\boldsymbol{A}_{ij} = 1$ if and only if $(j \rightarrow i) \in \mathcal{E}$, and let $\boldsymbol{D} := \text{diag}(\boldsymbol{A}\boldsymbol{1}_N)$ denote the in-degree matrix. The geodesic distance, $d_{\mathcal{G}}(j, i)$, from node $j$ to node $i$ is the length of the shortest path starting at node $j$ and ending at node $i$. Accordingly, the $\ell$-hop neighborhood of a node $i$ can be defined as the set of nodes that can reach it in exactly $\ell \in \mathbb{N}_0$ steps, $\mathbb{S}^{(\ell)}(i) = \{j \in \mathcal{V} : d_{\mathcal{G}}(j, i) = \ell\}$.

### 2.1 GRAPH NEURAL NETWORKS

Graph neural networks (GNNs) operate on inputs of the form $(\mathcal{G}, \boldsymbol{X})$, where $\mathcal{G}$ encodes the graph topology and $\boldsymbol{X} \in \mathbb{R}^{N \times H^{(0)}}$ collects the node features[2]. Message-passing neural networks (MPNNs) (Gilmer et al., 2017) are a special class of GNNs which recursively aggregate information

---

[1]Sometimes, along with removal of some edges to preserve statistical properties of the original topology.
[2]To keep things simple, we will ignore edge features.

from the 1-hop neighborhood of each node using *message-passing layers*. An L-layer MPNN is given as

$$z_i^{(\ell)} = \mathsf{Upd}^{(\ell)} \left( z_i^{(\ell-1)}, \mathsf{Agg}^{(\ell)} \left( z_i^{(\ell-1)}, \left\{ z_j^{(\ell-1)} : j \in \mathbb{S}^{(1)}(i) \right\} \right) \right), \quad \forall \ell \in [L]$$

$$\mathsf{MPNN}_\theta(\mathcal{G}, \boldsymbol{X}) = \left\{ \mathsf{Out} \left( z_i^{(L)} \right) : i \in \mathcal{V} \right\} \tag{2.1}$$

where $\boldsymbol{Z}^{(0)} = \boldsymbol{X}$, $\mathsf{Agg}^{(\ell)}$ denotes the *aggregation functions*, $\mathsf{Upd}^{(\ell)}$ the *update functions*, and Out the *readout function*. Since the final representation of node $i$ is a function of the input features of nodes at most L-hops away from it, its *receptive field* is given by $\mathbb{B}^{(L)}(i) := \{j \in \mathcal{V} : d_\mathcal{G}(j, i) \le L\}$.

For example, a GCN (Kipf & Welling, 2017) updates node representations as the weighted sum of its neighbors' representations:

$$\boldsymbol{Z}^{(\ell)} = \sigma \left( \hat{\boldsymbol{A}} \boldsymbol{Z}^{(\ell-1)} \boldsymbol{W}^{(\ell)} \right) \tag{2.2}$$

where $\sigma$ is a point-wise nonlinearity, e.g. ReLU, the propagation matrix, $\hat{\boldsymbol{A}}$, is a *graph shift operator*, i.e. $\hat{\boldsymbol{A}}_{ij} \ne 0$ if and only if $(j \to i) \in \mathcal{E}$ or $i = j$, and $\boldsymbol{W}^{(\ell)} \in \mathbb{R}^{H^{(\ell-1)} \times H^{(\ell)}}$ is a weight matrix. The original choice for $\hat{\boldsymbol{A}}$ was the symmetrically normalized adjacency matrix $\hat{\boldsymbol{A}}^{\text{sym}} := \widetilde{\boldsymbol{D}}^{-1/2} \widetilde{\boldsymbol{A}} \widetilde{\boldsymbol{D}}^{-1/2}$ (Kipf & Welling, 2017), where $\widetilde{\boldsymbol{A}} = \boldsymbol{A} + \boldsymbol{I}_N$ and $\widetilde{\boldsymbol{D}} = \text{diag}(\widetilde{\boldsymbol{A}} \mathbf{1}_N)$. However, several influential works have also used the asymmetrically normalized adjacency, $\hat{\boldsymbol{A}}^{\text{asym}} := \widetilde{\boldsymbol{D}}^{-1} \widetilde{\boldsymbol{A}}$ (Hamilton et al., 2017; Li et al., 2018; Schlichtkrull et al., 2017).

## 2.2 DROPEDGE

DropEdge is a random data augmentation technique that works by sampling a subgraph of the original input graph in each layer, and uses that for message passing (Rong et al., 2020):

$$\mathbf{M}^{(\ell)} \sim \{\text{Bern}(1-q)\}^{N \times N}$$

$$\widetilde{\boldsymbol{A}}^{(\ell)} = \mathbf{M}^{(\ell)} \circ \boldsymbol{A} + \boldsymbol{I}_N \tag{2.3}$$

Several variants of DropEdge have also been proposed, forming a family of random edge-dropping algorithms for tackling the over-smoothing problem. For example, DropNode (Feng et al., 2020) independently samples nodes and sets their representations to 0, followed by rescaling to make the feature matrix unbiased. This is equivalent to setting the corresponding columns of the propagation matrix to 0. In a similar vein, DropAgg (Jiang et al., 2023) samples nodes that don't aggregate messages from their neighbors. This is equivalent to dropping the corresponding rows of the adjacency matrix. Combining these two approaches, DropGNN (Papp et al., 2021) samples nodes which neither propagate nor aggregate messages in a given message-passing step. These algorithms alleviate over-smoothing by reducing the number of messages being propagated in the graph, thereby slowing down the convergence of node representations.

## 2.3 OVER-SQUASHING

Over-squashing refers to the problem of information from exponentially growing neighborhoods (Chen et al., 2018a) being squashed into finite-sized node representations (Alon & Yahav, 2021). This results in a loss of information as it is propagated over long distances, disallowing MPNNs from capturing long-range interactions (LRIs) and limiting their applications to short-range tasks. Topping et al. (2022) formally characterized over-squashing in terms of the Jacobian of the node-level representations w.r.t. the input features: $\|\partial z_i^{(L)}/\partial \boldsymbol{x}_j\|_1$. Accordingly, over-squashing can be understood as low sensitivity between distant nodes, i.e. small perturbations in a node's features don't effect other distant nodes' representations.

Several works have linked over-squashing in an MPNN with topological properties like Cheeger's constant (Giraldo et al., 2023; Karhadkar et al., 2023), curvature of edges (Liu et al., 2023; Nguyen et al., 2023; Topping et al., 2022), effective resistance between nodes (Arnaiz-Rodríguez et al., 2022; Black et al., 2023) and the expected commute time between them (Di Giovanni et al., 2023; Giovanni et al., 2024). These results have inspired the design of several graph rewiring techniques that strategically add edges to improve the connectivity in the graph, thereby alleviating over-squashing.

## 3 SENSITIVITY ANALYSIS

In this section, we perform a theoretical analysis of the expectation – w.r.t. random edge masks – of sensitivity of node representations. This will allow us to predict how DropEdge-variants affect communication between nodes at various distances, which is relevant for predicting their suitability towards learning LRIs.

### 3.1 LINEAR GCNS

We start our analysis with linear GCNs, and treat more general MPNN architectures in the following subsection. In this model, the final node representations can be summarised as

$$\boldsymbol{Z}^{(L)} = \left(\prod_{\ell=1}^{L} \hat{\boldsymbol{A}}^{(\ell)}\right) \boldsymbol{X} \boldsymbol{W} \in \mathbb{R}^{N \times H^{(L)}} \tag{3.1}$$

where $\boldsymbol{W} \coloneqq \prod_{\ell=1}^{L} \boldsymbol{W}^{(\ell)} \in \mathbb{R}^{H^{(0)} \times H^{(L)}}$. Using the i.i.d. assumption on the distribution of edge masks in each layer, the expected sensitivity of node $i$ to node $j$ can be shown to be

$$\mathbb{E}_{\mathbf{M}^{(1)}, \dots, \mathbf{M}^{(L)}} \left[\left\|\frac{\partial \boldsymbol{z}_i^{(L)}}{\partial \boldsymbol{x}_j}\right\|_1\right] = \left(\mathbb{E}\left[\hat{\boldsymbol{A}}\right]^L\right)_{ij} \|\boldsymbol{W}\|_1 \tag{3.2}$$

To keep things simple, we will ignore the effect of DropEdge-variants on the optimization trajectory. Accordingly, it is sufficient to study $\mathbb{E}[\hat{\boldsymbol{A}}]$ in order to predict their effect on over-squashing. To maintain analytical tractability, we assume the use of an asymmetrically normalized adjacency matrix for message-passing, $\hat{\boldsymbol{A}} = \hat{\boldsymbol{A}}^{\text{asym}}$.

**Lemma 3.1.** *The expected propagation matrix under DropEdge is given as:*

$$\begin{aligned} \dot{\boldsymbol{P}}_{ii} &\coloneqq \mathbb{E}_{\text{DE}}\left[\hat{\boldsymbol{A}}_{ii}\right] = \frac{1 - q^{d_i+1}}{(1-q)(d_i+1)} \\ \dot{\boldsymbol{P}}_{ij} &\coloneqq \mathbb{E}_{\text{DE}}\left[\hat{\boldsymbol{A}}_{ij}\right] = \frac{1}{d_i}\left(1 - \frac{1 - q^{d_i+1}}{(1-q)(d_i+1)}\right) \end{aligned} \tag{3.3}$$

*where $q \in [0, 1)$ is the dropping probability.*

See Appendix B.1 for a proof.

**Other Variants**. We will similarly derive the expected propagation matrix for other random edge-dropping algorithms. First off, DropNode (Feng et al., 2020) samples nodes and drops corresponding columns from the aggregation matrix directly, followed by rescaling of its entries:

$$\mathbb{E}_{\text{DN}}\left[\frac{1}{1-q}\hat{\boldsymbol{A}}\right] = \frac{1}{1-q} \times (1-q)\,\hat{\boldsymbol{A}} = \hat{\boldsymbol{A}} \tag{3.4}$$

That is, the expected propagation matrix is the same as in a NoDrop model ($q = 0$).

Nodes sampled by DropAgg (Jiang et al., 2023) don't aggregate messages. Therefore, if $\hat{\boldsymbol{A}} = \hat{\boldsymbol{A}}^{\text{asym}}$, then the expected propagation matrix is given by

$$\begin{aligned} \mathbb{E}_{\text{DA}}\left[\hat{\boldsymbol{A}}_{ii}\right] &= q + \frac{1-q}{d_i+1} = \frac{1+d_iq}{d_i+1} > \mathbb{E}_{\text{DE}}\left[\hat{\boldsymbol{A}}_{ii}\right] \\ \mathbb{E}_{\text{DA}}\left[\hat{\boldsymbol{A}}_{ij}\right] &= \frac{1}{d_i}\left(1 - \mathbb{E}_{\text{DA}}\left[\hat{\boldsymbol{A}}_{ii}\right]\right) < \mathbb{E}_{\text{DE}}\left[\hat{\boldsymbol{A}}_{ij}\right] \end{aligned} \tag{3.5}$$

Finally, DropGNN (Papp et al., 2021) samples nodes which neither propagate nor aggregate messages. From any node's perspective, if it is not sampled, then its aggregation weights are computed

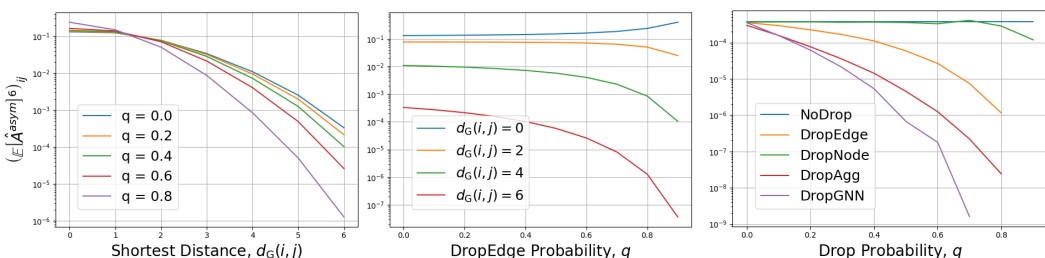

**Figure 1:** Entries of $\dot{\boldsymbol{P}}^6$, averaged over molecular graphs sampled from the Proteins dataset. *Left*: Sensitivity between nodes decays at a polynomial rate w.r.t. their distance. *Middle*: Similarly, it decays at a polynomial rate w.r.t. the DropEdge probability. *Right*: It decays more quickly with DropAgg than with DropEdge, and even more rapidly with DropGNN.

as for DropEdge:

$$
\begin{aligned}
\mathbb{E}_{\text{DG}}\left[\hat{\boldsymbol{A}}_{ii}\right] &= q + (1-q)\,\mathbb{E}_{\text{DE}}\left[\hat{\boldsymbol{A}}_{ii}\right] = q + \frac{1 - q^{d_i+1}}{d_i + 1} > \mathbb{E}_{\text{DA}}\left[\hat{\boldsymbol{A}}_{ii}\right] \\
\mathbb{E}_{\text{DG}}\left[\hat{\boldsymbol{A}}_{ij}\right] &= \frac{1}{d_i}\left(1 - \mathbb{E}_{\text{DG}}\left[\hat{\boldsymbol{A}}_{ii}\right]\right) < \mathbb{E}_{\text{DA}}\left[\hat{\boldsymbol{A}}_{ij}\right]
\end{aligned}
\tag{3.6}
$$

**1-Layer Linear GCNs**. $\forall q \in (0,1)$ we have

$$
\begin{aligned}
\dot{\boldsymbol{P}}_{ii} &= \frac{1}{d_i + 1}\sum_{k=0}^{d_i} q^k > \frac{1}{d_i + 1} \\
\dot{\boldsymbol{P}}_{ij} &= \frac{1}{d_i}\left(1 - \dot{\boldsymbol{P}}_{ii}\right) < \frac{1}{d_i + 1}
\end{aligned}
\tag{3.7}
$$

where the right-hand sides of the two inequalities are the corresponding entries in the propagation matrix of a NoDrop model. Equation 3.3 to Equation 3.7 together imply the following result:

**Lemma 3.2.** *In a 1-layer linear GCN with $\hat{\boldsymbol{A}} = \hat{\boldsymbol{A}}^{asym}$, using DropEdge, DropAgg or DropGNN*

1. *increases the sensitivity of a node's representations to its own input features, and*

2. *decreases the sensitivity to its neighbors' features.*

In other words, DropEdge-variants prevent a 1-layer GCN from fully utilizing neighborhood information when learning node representations. Ignoring the graph topology this way makes the model resemble an MLP, limiting its expressiveness and hindering its ability to model graph-data.

**L-layer Linear GCNs**. Unfortunately, we cannot draw similar conclusions in L-layer networks, for nodes at arbitrary distances. To see this, view $\dot{\boldsymbol{P}}$ as the transition matrix of a non-uniform random walk. This walk has higher self-transition ($i = j$) probabilities than in a uniform augmented random walk ($\boldsymbol{P} = \hat{\boldsymbol{A}}^{\text{asym}}$, $q = 0$), but lower inter-node ($i \neq j$) transition probabilities. Note that $\dot{\boldsymbol{P}}^L$ and $\boldsymbol{P}^L$ store the L-step transition probabilities in the corresponding walks. Then, since the paths connecting the nodes $i \in \mathcal{V}$ and $j \in \mathbb{B}^{(L-1)}(i)$ may involve self-loops, $(\dot{\boldsymbol{P}}^L)_{ij}$ may be lower or higher than $(\boldsymbol{P}^L)_{ij}$. Therefore, we cannot conclude how sensitivity between nodes separated by at most $L-1$ hops changes. For nodes L-hops away, however, we can show that DropEdge always decreases the corresponding entry in $\dot{\boldsymbol{P}}^L$, reducing the effective reachability of GCNs. Using Equation 3.5 and Equation 3.6, we can show the same for DropAgg and DropGNN, respectively.

**Theorem 3.1.** *In an L-layer linear GCN with $\hat{\boldsymbol{A}} = \hat{\boldsymbol{A}}^{asym}$, using DropEdge, DropAgg or DropGNN decreases the sensitivity of a node $i \in \mathcal{V}$ to another node $j \in \mathbb{S}^{(L)}(i)$, thereby reducing its effective receptive field. Moreover, the sensitivity monotonically decreases as the dropping probability is increased.*

See Appendix B.2 for a precise quantitative statement and the proof.

**Nodes at Arbitrary Distances**. Although no general statement could be made about the change in sensitivity between nodes up to $L-1$ hops away, we can analyze such pairs empirically. We sampled 100 molecular graphs from the Proteins dataset (Dobson & Doig, 2003), binned the node-pairs in each graph by the shortest distance between them, and then plotted the average of the corresponding entries in $\dot{\boldsymbol{P}}^L$, $L = 6$.

The results are shown in Figure 1. In the left subfigure, we observe that the sensitivity between two nodes decays at a *polynomial* rate with increasing distance between them. Moreover, DropEdge increases the expected sensitivity between nodes close to each other (0-hop and 1-hop neighbors) in the original topology, but reduces it between nodes farther off. In the middle subfigure, we show how the average sensitivity at different distances changes with the DropEdge probability. Specifically, we observe that the decay in sensitivity to nodes at large distances is polynomial in the DropEdge probability, which suggests that the algorithm would not be suitable for capturing LRIs. Similar conclusions can be made with the symmetrically normalized propagation matrix (see Appendix D.1). Finally, in the right subfigure, we compare the DropEdge-variants by plotting the sensitivity at $d_{\mathcal{G}}(i, j) = L = 6$. Although the analytical form of all the expected propagation matrices are available to us, we approximate them using Monte-Carlo sampling. As expected from Equation 3.5, we observe that DropAgg not only decreases the sensitivity of node representations to distant nodes, but does it to a greater extent than DropEdge. Similarly, Equation 3.6 suggested that DropGNN could be even more harmful than DropAgg, and this is validated by the empirical results. In fact, for $q > 0.7$, nodes 6-hops away are mostly insensitive to each other.

## 3.2 Nonlinear MPNNs

While linear networks are useful in simplifying the theoretical analysis, they are often not practical. In this subsection, we will consider the upper bounds on sensitivity established in previous works, and extend them to the DropEdge setting.

**ReLU GCNs**. Xu et al. (2018) considered the case of ReLU nonlinearity, so that the update rule is $\boldsymbol{Z}^{(\ell)} = \mathsf{ReLU}(\hat{\boldsymbol{A}}\boldsymbol{Z}^{(\ell-1)}\boldsymbol{W}^{(\ell)})$. Additionally, it makes the simplifying assumption that each path in the computational graph is *active* with a fixed probability, $\rho$ (Kawaguchi, 2016, Assumption A1p-m). Accordingly, the sensitivity (in expectation) between any two nodes is given as

$$\left\| \mathbb{E}_{\mathsf{ReLU}}\left[ \frac{\partial \boldsymbol{z}_i^{(L)}}{\partial \boldsymbol{x}_j} \right] \right\|_1 = \left[ \rho \left\| \prod_{\ell=1}^{L} \boldsymbol{W}^{(\ell)} \right\|_1 \right] \left( \hat{\boldsymbol{A}}^L \right)_{ij} = \zeta_1^{(L)} \left( \hat{\boldsymbol{A}}^L \right)_{ij} \tag{3.8}$$

where $\zeta_1^{(L)}$ is independent of the choice of nodes $i, j \in \mathcal{V}$. Taking an expectation w.r.t. the random edge masks, we get

$$\mathbb{E}_{\mathbf{M}^{(1)},\dots,\mathbf{M}^{(L)}}\left[ \left\| \mathbb{E}_{\mathsf{ReLU}}\left[ \frac{\partial \boldsymbol{z}_i^{(L)}}{\partial \boldsymbol{x}_j} \right] \right\|_1 \right] = \zeta_1^{(L)} \mathbb{E}_{\mathbf{M}^{(1)},\dots,\mathbf{M}^{(L)}}\left[ \left( \prod_{\ell=1}^{L} \hat{\boldsymbol{A}}^{(\ell)} \right)_{ij} \right] \tag{3.9}$$

$$= \zeta_1^{(L)} \left( \mathbb{E}\left[ \hat{\boldsymbol{A}} \right]^L \right)_{ij} \tag{3.10}$$

Using Theorem 3.1, we conclude that in a ReLU-GCN, DropEdge, DropAgg and DropGNN will reduce the expected sensitivity between nodes L-hops away. Empirical observations in Figure 1 and Figure 4 suggest that we may expect an increase in sensitivity to neighboring nodes, but a significant decrease in sensitivity to those farther away.

**Source-only Message Functions**. Black et al. (2023, Lemma 3.2) considers MPNNs with aggregation functions of the form

$$\mathsf{Agg}^{(\ell)}\left( \boldsymbol{z}_i^{(\ell-1)}, \left\{ \boldsymbol{z}_j^{(\ell-1)} : j \in \mathbb{S}^{(1)}(i) \right\} \right) = \sum_{j \in \mathbb{B}^{(1)}(i)} \hat{\boldsymbol{A}}_{ij} \mathsf{Msg}^{(\ell)}\left( \boldsymbol{z}_j^{(\ell-1)} \right) \tag{3.11}$$

and Upd and Msg functions with bounded gradients. In this case, the sensitivity between two nodes $i, j \in \mathcal{V}$ can be bounded as

$$\left\| \frac{\partial \boldsymbol{z}_i^{(L)}}{\partial \boldsymbol{x}_j} \right\|_1 \le \zeta_2^{(L)} \left( \sum_{\ell=0}^{L} \hat{\boldsymbol{A}}^\ell \right)_{ij} \tag{3.12}$$

As before, we can use the independence of edge masks to get an upper bound on the expected sensitivity:

$$\mathbb{E}_{\mathbf{M}^{(1)},\ldots,\mathbf{M}^{(L)}} \left[ \left\| \frac{\partial \boldsymbol{z}_i^{(L)}}{\partial \boldsymbol{x}_j} \right\|_1 \right] \leq \zeta_2^{(L)} \left( \mathbb{E}_{\mathbf{M}^{(1)},\ldots,\mathbf{M}^{(L)}} \left[ I_N + \sum_{\ell=1}^{L} \prod_{k=1}^{\ell} \hat{\boldsymbol{A}}^{(k)} \right] \right)_{ij} \tag{3.13}$$

$$= \zeta_2^{(L)} \left( \sum_{\ell=0}^{L} \mathbb{E}\left[\hat{\boldsymbol{A}}\right]^{\ell} \right)_{ij} \tag{3.14}$$

Figure 5 shows the plot of the entries of $\sum_{\ell=0}^{6} \dot{\boldsymbol{P}}^{\ell}$ (i.e. for DropEdge), as in the upper bound above, with $\hat{\boldsymbol{A}} = \hat{\boldsymbol{A}}^{\text{asym}}$. We observe that the sensitivity between nearby nodes marginally increases, while that between distant nodes notably decreases (similar to Figure 1), suggesting significant over-squashing. Similar observations can be made with $\hat{\boldsymbol{A}} = \hat{\boldsymbol{A}}^{\text{sym}}$, and for other DropEdge-variants.

**Source-and-Target Message Functions**. Topping et al. (2022, Lemma 1) showed that if the aggregation function is instead given by

$$\text{Agg}^{(\ell)} \left( \boldsymbol{z}_i^{(\ell-1)}, \left\{ \boldsymbol{z}_j^{(\ell-1)} : j \in \mathbb{S}^{(1)}(i) \right\} \right) = \sum_{j \in \mathbb{B}^{(1)}(i)} \hat{\boldsymbol{A}}_{ij} \text{Msg}^{(\ell)} \left( \boldsymbol{z}_i^{(\ell-1)}, \boldsymbol{z}_j^{(\ell-1)} \right) \tag{3.15}$$

then the sensitivity between nodes $i \in \mathcal{V}$ and $j \in \mathbb{S}^{(L)}(i)$ can be bounded as

$$\left\| \frac{\partial \boldsymbol{z}_i^{(L)}}{\partial \boldsymbol{x}_j} \right\|_1 \leq \zeta_3^{(L)} \left( \hat{\boldsymbol{A}}^L \right)_{ij} \tag{3.16}$$

With random edge-dropping, this bound can be adapted as follows:

$$\mathbb{E}_{\mathbf{M}^{(1)},\ldots,\mathbf{M}^{(L)}} \left[ \left\| \frac{\partial \boldsymbol{z}_i^{(L)}}{\partial \boldsymbol{x}_j} \right\|_1 \right] \leq \zeta_3^{(L)} \left( \mathbb{E}\left[\hat{\boldsymbol{A}}\right]^L \right)_{ij} \tag{3.17}$$

which is similar to Equation 3.10, only with a different proportionality constant, that is anyway independent of the choice of nodes. Here, again, we invoke Theorem 3.1 to conclude that $(\mathbb{E}[\hat{\boldsymbol{A}}]^L)_{ij}$ decreases monotonically with increasing DropEdge probability $q$. This implies that, in a non-linear MPNN with $\hat{\boldsymbol{A}} = \hat{\boldsymbol{A}}^{\text{asym}}$, DropEdge lowers the sensitivity bound given above. Empirical results in Figure 4 support the same conclusion for $\hat{\boldsymbol{A}} = \hat{\boldsymbol{A}}^{\text{sym}}$.

> **Message of the Section**: Studying the expected propagation matrix allows us to predict the effect of random edge-dropping methods on information propagation in MPNNs. Specifically, DropEdge, DropAgg and DropGNN increase the sensitivity of nodes to their neighbors, but decrease it to nodes farther off. This suggests that these methods would be unsuitable for long-range tasks, where it is imperative to facilitate communication between distant nodes.

## 4 EXPERIMENTS

Our theoretical analysis indicates that DropEdge may degrade the performance of GNNs on tasks that depend on capturing LRIs. In this section, we test this hypothesis by evaluating DropEdge models on both short-range and long-range tasks.

### 4.1 SETUP

**Datasets**. Although identifying whether a task requires modeling LRIs can be challenging, understanding the structure of the datasets can provide some insight. For example, homophilic datasets have local consistency in node labels, i.e. nodes closely connected to each other have similar labels. On the other hand, in heterophilic datasets, nearby nodes often have dissimilar labels. Since DropEdge-variants increase a node's sensitivity to its immediate neighbors and reduces its sensitivity

| Dataset | Nodes | Edges | Features | Classes | Homophily |
|---------|-------|-------|----------|---------|-----------|
| Cora | 2,708 | 10,556 | 1,433 | 7 | 0.766 |
| CiteSeer | 3,327 | 9,104 | 3,703 | 6 | 0.627 |
| Chameleon | 2,277 | 36,051 | 2,325 | 5 | 0.062 |
| Squirrel | 5,201 | 216,933 | 2,089 | 5 | 0.025 |
| TwitchDE | 9,498 | 306,276 | 128 | 2 | 0.142 |

**Table 1:** Dataset statistics. Number of edges excludes self-loops. Homophily measures from (Lim et al., 2021).

| Dropout | GNN | Homophilic | | Heterophilic | | |
|---------|-----|------------|----------|--------------|----------|----------|
| | | Cora | CiteSeer | Chameleon | Squirrel | TwitchDE |
| DropEdge | GCN | +0.396 | +0.448 | −0.723 | −0.614 | −0.509 |
| | GAT | +0.547 | +0.548 | −0.236 | −0.655 | −0.409 |
| DropNode | GCN | −0.320 | −0.662 | −0.863 | −0.790 | −0.465 |
| | GAT | −0.706 | −0.735 | −0.855 | −0.300 | −0.502 |
| DropAgg | GCN | −0.015 | +0.294 | −0.334 | −0.317 | −0.566 |
| | GAT | −0.019 | +0.392 | −0.353 | −0.398 | −0.022 |
| DropGNN | GCN | +0.360 | +0.468 | −0.746 | −0.600 | −0.671 |
| | GAT | +0.236 | +0.525 | −0.507 | −0.304 | −0.139 |

**Table 2:** Spearman correlation between dropping probability and test accuracy. Note the positive correlations for homophilic datasets and negative correlations for heterophilic datasets. Results suggest that random edge-dropping, although effective at improving generalization in short-range tasks, is unsuitable for long-range ones.

to distant nodes, we expect it to improve performance on homophilic datasets but harm performance on heterophilic ones. In this work, we use Cora (McCallum et al., 2000) and CiteSeer (Giles et al., 1998) as representatives of homophilic datasets (Zhu et al., 2020), and Squirrel, Chameleon and TwitchDE (Rozemberczki et al., 2021) to represent heterophilic datasets (Lim et al., 2021). The networks' statistics are presented in Table 1, where we can note the significantly lower homophily measures of heterophilic datasets.

**Models**. We use two MPNN architectures, GCN and GAT, to demonstrate the effect of DropEdge. GCN satisfies the model assumptions made in all the theoretical results presented in Section 3, while GAT does not fit into any of them. Therefore, GCN and GAT together provide a broad representation of different MPNN architectures. For GAT, we use 2 attentions heads in order to keep the computational load manageable, while at the same time harnessing the expressiveness of the multi-headed self-attention mechanism. As for the model depth, we vary it as $L = 2, 4, 6, 8$.

**Dropping Probability**. For all the methods, the dropping probabilities are varied as $q = 0.0, 0.1, \ldots, 0.9$, so as to reliably capture the trends in model accuracy. We adopt the common practice of turning the methods off at test-time ($q = 0$), isolating their effect on optimization and generalization, which our theory does not address.

Other experimental details are reported in Appendix E. We conduct 5 independent runs for each dataset−model−dropout−probability configuration and report the average test accuracy (with early-stopping using the validation set).

## 4.2 RESULTS

In Table 2, we report the rank correlation between the dropping probability and test accuracy of different dataset−model−dropout combinations. The statistics are computed over 10 ($q = 0.0, 0.1, \ldots, 0.9$) × 5 (samples) = 50 runs for each $L = 2, 4, 6, 8$, and then averaged. It is clear to see that in most combinations with the homophilic datasets Cora and CiteSeer, the correlation is positive, indicating that DropEdge and its variants improve test-time performance at short-range tasks. On the other hand, with the heterophilic datasets Chameleon, Squirrel and TwitchDE, the

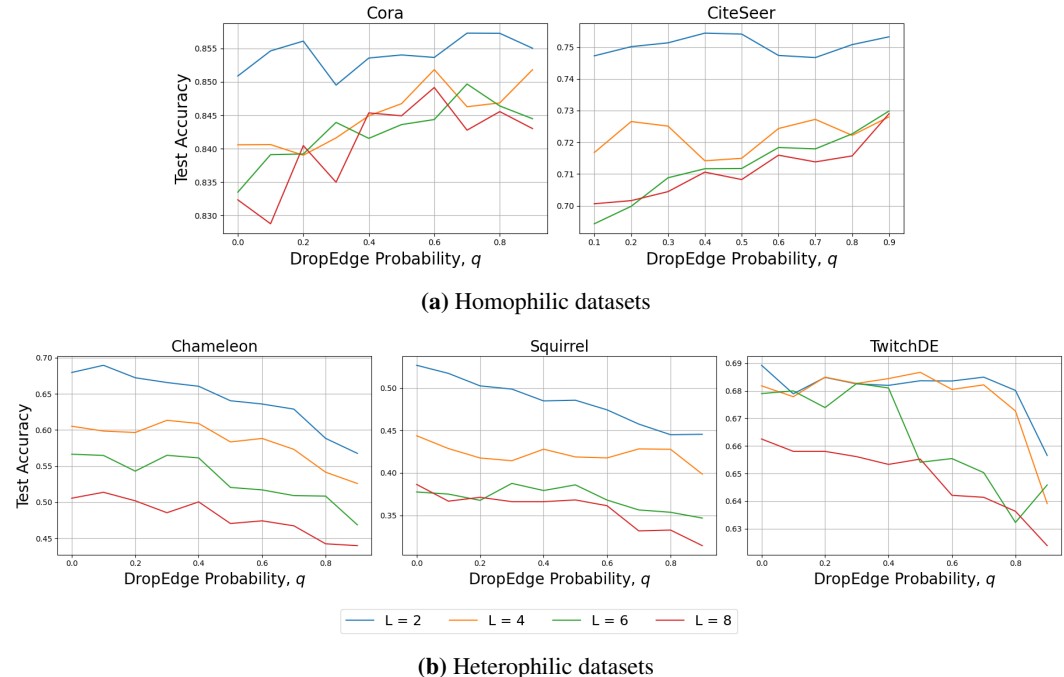

**(a)** Homophilic datasets

**(b)** Heterophilic datasets

**Figure 2:** Dropping probability versus test accuracy of DropEdge-GCN. The theory the explains the contrasting trends as follows: random edge-dropping pushes models to fit to local information during training, which is suitable for short-range tasks, but harms test-time performance in long-range ones.

correlation values are negative. This suggests that the edge-dropping methods harm generalization in long-range tasks by forcing the model to overfit to short-range signals.

Figure 2 allows us to visualize these trends for the homophilic and heterophilic datasets. For conciseness, we only display the results for DropEdge-GCNs. Clearly, on Cora and CiteSeer, the model's test-time performance improves with increasing dropping probability, as has been reported earlier. However, on Chameleon, Squirrel and TwitchDE, the performance degrades with increasing dropping probability, as was suggested by our theoretical results. This highlights an important gap in our understanding of dropout methods – while their positive effects on model performance have been well-studied, making them popular choices for training deep GNNs, their evaluation has been limited to short-range tasks, leaving their negative impact on capturing LRIs overlooked.

**Remark on DropNode**. In Equation 3.4, we noted that DropNode does not suffer from loss in sensitivity. However, note that those results were in expectation. Moreover, our analysis did not account for the effects on the learning trajectory. In practice, a high DropNode probability would make it hard for information in the node features to reach distant nodes. This would prevent the model from learning to effectively combine information from large neighborhoods, harming generalization. In fact, in Table 2, we can see that it is the only dropping method with a negative correlation between dropping probability and test accuracy for each dataset−model combination, including homophilic ones. See Figure 6 for a visualization of its negative effects on test accuracy.

### 4.3 OVER-SQUASHING OR UNDER-FITTING?

The results in the previous subsection suggest that using random edge-dropping to regularize model training leads to poor test-time performance. We hypothesize that this occurs because the models struggle to propagate information over long distances, causing overfitting of node representations to local neighborhoods. However, a confounding factor is at play: DropEdge variants reduce the generalization gap by preventing overfitting to the training set, which manifests as reduced training performance. If this regularization is too strong, it could lead to underfitting, which could also explain the poor test-time performance on heterophilic datasets. This concern is particularly relevant because the heterophilic networks are much larger than homophilic ones (Table 1), making them

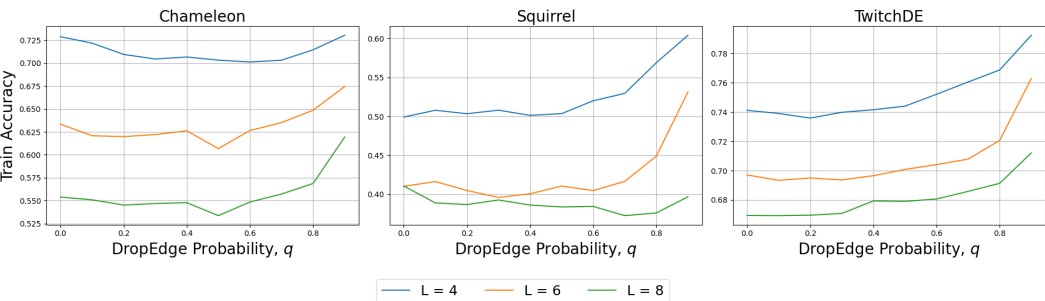

**Figure 3:** DropEdge probability versus training accuracy of GCNs. The training performance improves with $q$, suggesting that the models are not underfitting. Instead, the reason for poor test-time performance (Figure 2a) is that models are over-fitting to short-range signals during training, resulting in poor generalization.

more prone to underfitting. To investigate this, we plot the training accuracies of deep DropEdge-GCNs on the heterophilic datasets; Figure 3 shows the results. It is clear that the models do not underfit as the dropping probability increases. In fact, somewhat unexpectedly, the training metrics improve. Together with the results in Figure 2, we conclude that DropEdge-like methods are detrimental in long-range tasks, as they cause overfitting to short-range artifacts in the training data, resulting in poor generalization at test-time.

> **Message of the Section**: Random edge-dropping algorithms reduce the receptive field of MPNNs, forcing them to fit to short-range signals in the training set. While this may make deep GNNs suitable for homophilic datasets, it results in overfitting on heterophilic datasets, which leads to poor test-time performance. Therefore, these methods should be used only after carefully understanding the task at hand.

## 5 CONCLUSION

Our analysis points out a key assumption in algorithms designed for training deep GNNs: the idea that if a deep GNN is trainable, it must be able to model LRIs. Our results suggest that this, in fact, need not be true – we theoretically and empirically show that DropEdge-like algorithms exacerbate the over-squashing problem in deep GNNs, and degrade their performance on long-range tasks. Our results highlight a need for a thorough evaluation of methods employed when training deep GNNs, with regards to their capacity to capture LRIs. This will allow us to reliably deploy them in real-life, since we can be assured that the models did not simply overfit to short-range signals.

**Limitations**. While our theoretical analysis successfully predicts how DropEdge-variants affect test performance on short-range and long-range tasks, it is based on several simplifying assumptions. These assumptions, although standard in the literature, limit the generalizability of our conclusions to other architectures. Specifically, our analysis focuses on certain classes of MPNNs, excluding several GNN architectures specifically designed to enhance long-distance information propagation (see Singh (2024) for a review).

**Practical Considerations**. Previous studies have shown that random edge-dropping algorithms can effectively enhance generalization performance in short-range tasks, and our findings support this conclusion. However, we have also demonstrated that such algorithms can negatively impact over-squashing in MPNNs and harm test-time performance in long-range tasks. Therefore, we recommend exercising caution when using such methods, as careless application can result in models that generalize poorly, which can be detrimental in critical applications.

**Future Directions**. This work focuses on node-classification tasks, but it is also important to understand the effect of random edge-dropping in other practical settings, e.g. link prediction, and even graph-level tasks. In general, there is a need for a broader investigation into methods designed for training deep GNNs. Specifically, analyzing various strategies designed for mitigating over-smoothing (see Rusch et al. (2023) for a review), particularly in the context of over-squashing, could be invaluable for designing deep GNNs for long-range tasks.

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

# A  RELATED WORKS

## A.1  METHODS FOR ALLEVIATING OVER-SMOOTHING

A popular choice for reducing over-smoothing in GNNs is to regularize the model. Recall that DropEdge (Rong et al., 2020) implicitly regularizes the model by adding noise to the learning trajectory (Section 2.2). Similarly, Dropout (Srivastava et al., 2014) drops entries from the input features to each layer, and DropMessage (Fang et al., 2023) drops entries from the message matrix (before the aggregation step). Graph Drop Connect (GDC) (Hasanzadeh et al., 2020) combines DropEdge and DropMessage together, resulting in a layer-wise sampling scheme that uses a different sub-graph for message-aggregation over each feature dimension. These methods successfully addressed the over-smoothing problem, enabling the training of deep GNNs, and performed competitively on several benchmarking datasets.

Another powerful form of implicit regularization is feature normalization, which has proven crucial in enhancing the performance and stability of several types of neural networks (Huang et al., 2020). Exploiting the inductive bias in graph-structured data, normalization techniques like PairNorm (Zhao & Akoglu, 2020), Differentiable Group Normalization (DGN) (Zhou et al., 2020) and Node-Norm (Zhou et al., 2021b) have been proposed to reduce over-smoothing in GNNs. On the other hand, Energetic Graph Neural Networks (EGNNs) (Zhou et al., 2021a) explicitly regularize the optimization by constraining the layer-wise Dirichlet energy to a predefined range.

In a different vein, motivated by the success of residual networks (ResNets) (He et al., 2016) in computer vision, (Li et al., 2019) proposed the use of residual connections to prevent the smoothing of representations. Residual connections successfully improved the performance of GCN on a range of graph-learning tasks. (Chen et al., 2020) introduced GCNII, which uses skip connections from the input to all hidden layers. This layer wise propagation rule has allowed for training of ultra-deep networks – up to 64 layers. Some other architectures, like the Jumping Knowledge Network (JKNet) (Xu et al., 2018) and the Deep Adaptive GNN (DAGNN) (Liu et al., 2020), aggregate the representations from *all* layers, $\{z_i^{(\ell)}\}_{\ell=1}^{L}$, before processing them through a readout layer.

## A.2  METHODS FOR ALLEVIATING OVER-SQUASHING

In this section, we will review some of the graph rewiring methods proposed to address the problem of over-squashing. Particularly, we wish to emphasize a commonality among these methods – edge addition is necessary. As a reminder, *graph rewiring* refers to modifying the edge set of a graph by adding and/or removing edges in a systematic manner. In a special case, which includes many of the rewiring techniques we will discuss, the original topology is completely discarded, and only the rewired graph is used for message-passing.

*Spatial* rewiring methods use the topological relationships between the nodes in order to come up with a rewiring strategy. That is the graph rewiring is guided by the objective of optimizing some chosen topological properties. For instance, Alon & Yahav (2021) introduced a fully-adjacent (FA) layer, wherein messages are passed between all nodes. GNNs using a FA layer in the final message-passing step were shown to outperform the baselines on a variety of long-range tasks, revealing the importance of information exchange between far-off nodes which standard message-passing cannot facilitate. Topping et al. (2022) proposed a curvature-based rewiring strategy, called the Stochastic Discrete Ricci Flow (SDRF), which aims to reduce the "bottleneckedness" of a graph by adding suitable edges, while simultaneously removing edges in an effort to preserve the statistical properties of the original topology. Black et al. (2023) proposed the Greedy Total Resistance (GTR) technique, which optimizes the graph's total resistance by greedily adding edges to achieve the greatest improvement. One concern with graph rewiring methods is that unmoderated densification of the graph, e.g. using a fully connected graph for propagating messages, can result in a loss of the inductive bias the topology provides, potentially leading to over-fitting. Accordingly, Gutteridge et al. (2023) propose a Dynamically Rewired (DRew) message-passing framework that gradually *densifies* the graph. Specifically, in a given layer $\ell$, node $i$ aggregates messages from its entire $\ell$-hop receptive field instead of just the immediate neighbors. This results in an improved communication over long distances while also retaining the inductive bias of the shortest distance between nodes.

*Spectral* methods, on the other hand, use the spectral properties of the matrices encoding the graph topology, e.g. the adjacency or the Laplacian matrix, to design rewiring algorithms. For example, Arnaiz-Rodríguez et al. (2022) proposed a differentiable graph rewiring layer based on the Lovász bound (Lovász, 1993, Corollary 3.3). Similarly, Banerjee et al. (2022) introduced the Random Local Edge Flip (RLEF) algorithm, which draws inspiration from the "Flip Markov Chain" (Feder et al., 2006; Mahlmann & Schindelhauer, 2005) – a sequence of such steps can convert a connected graph into an *expander graph* – a sparse graph with good connectivity (in terms of Cheeger's constant) – with high probability (Allen-Zhu et al., 2016; Cooper et al., 2019; Feder et al., 2006; Giakkoupis, 2022; Mahlmann & Schindelhauer, 2005), thereby enabling effective information propagation across the graph.

Some other rewiring techniques don't exactly classify as spatial or spectral methods. For instance, Probabilistically Rewired MPNN (PR-MPNN) (Qian et al., 2024) learns to probabilistically rewire a graph, effectively mitigating under-reaching as well as over-squashing. Finally, (Gabrielsson et al., 2023) proposed connecting all nodes at most $r$-hops away, for some $r \in \mathbb{N}$, and introducing positional embeddings to allow for distance-aware aggregation of messages.

### A.3 TOWARDS A UNIFIED TREATMENT

Several studies have shown that an inevitable trade-off exists between the problems of over-smoothing and over-squashing, meaning that optimizing for one will compromise the other. For instance, Nguyen et al. (2023); Topping et al. (2022) showed that negatively curved edges create bottlenecks in the graph resulting in over-squashing of information. On the other hand, Nguyen et al. (2023, Proposition 4.3) showed that positively curved edges in a graph contribute towards the over-smoothing problem. To address this trade-off, they proposed Batch Ollivier-Ricci Flow (BORF), which adds new edges adjacent to the negatively curved ones, and simultaneously removes positively curved ones. In a similar vein, Giraldo et al. (2023) demonstrated that the minimum number of message-passing steps required to reach a given level of over-smoothing is inversely related to the Cheeger's constant, $h_{\mathcal{G}}$. This again implies an inverse relationship between over-smoothing and over-squashing. To effectively alleviate the two issues together, they proposed the Stochastic Jost and Liu Curvature Rewiring (SJLR) algorithm, which adds edges that result in high improvement in the curvature of existing edges, while simultaneously removing those that have low curvature.

Despite the well-established trade-off between over-smoothing and over-squashing, some works have successfully tackled them together despite *only* adding or removing edges. One such work is Karhadkar et al. (2023), which proposed a rewiring algorithm that adds edges to the graph but does not remove any. The First-order Spectral Rewiring (FoSR) algorithm computes, as the name suggests, a first order approximation to the spectral gap of $\boldsymbol{L}^{\text{sym}}$, and adds edges with the aim of maximizing it. Since the spectral gap directly relates to Cheeger's constant through Cheeger's inequality (Alon & Milman, 1985; Alon, 1986; Sinclair & Jerrum, 1989), this directly decreases the over-squashing levels. Moreover, Karhadkar et al. (2023, Figure 5) empirically demonstrated that addition of (up to a small number of) edges selected by FoSR can lower the Dirichlet energy of the representations. Taking a completely opposite approach, CurvDrop (Liu et al., 2023) adapted DropEdge to remove negatively curved edges sampled from a distribution proportional to their curvatures. CurvDrop directly reduces over-squashing and, as a side benefit of operating on a sparser subgraph, also mitigates over-smoothing.

## B  PROOFS

### B.1  EXPECTED PROPAGATION MATRIX UNDER DROPEDGE

**Lemma.** *When using DropEdge, the expected propagation matrix is given as:*

$$\mathbb{E}_{\text{DE}}\left[\hat{\boldsymbol{A}}_{ii}^{(1)}\right] = \frac{1 - q^{d_i+1}}{(1-q)\,(d_i+1)}$$

$$\mathbb{E}_{\text{DE}}\left[\hat{\boldsymbol{A}}_{ij}^{(1)}\right] = \frac{1}{d_i}\left(1 - \frac{1 - q^{d_i+1}}{(1-q)\,(d_i+1)}\right)$$

*where $(j \rightarrow i) \in \mathcal{E}$; $\dot{\boldsymbol{P}}_{ij} = 0$ otherwise.*

*Proof.* Recall that under DropEdge, a self-loop is added to the graph *after* the edges are dropped, and then the normalization is performed. In other words, the self-loop is never dropped. Therefore, given the i.i.d. masks, $m_1, \ldots, m_{d_i} \sim \text{Bern}(1-q)$, on incoming edges to node $i$, the total number of messages is given by

$$1 + \sum_{k=1}^{d_i} m_k = 1 + M_i$$

where $M_i \sim \text{Binom}(d_i, 1-q)$. Under asymmetric normalization (see Section 2.1), the expected weight of the message along the self-loop is computed as follows:

$$\mathbb{E}_{\text{DE}}\left[\hat{\boldsymbol{A}}_{ii}^{(1)}\right] = \mathbb{E}_{m_1,\ldots,m_{d_i}}\left[\frac{1}{1 + \sum_{k=1}^{d_i} m_k}\right] \tag{B.1}$$

$$= \mathbb{E}_{M_i}\left[\frac{1}{1 + M_i}\right] \tag{B.2}$$

$$= \sum_{k=0}^{d_i} \binom{d_i}{k} (1-q)^k (q)^{d_i-k} \left(\frac{1}{1+k}\right) \tag{B.3}$$

$$= \frac{1}{(1-q)(d_i+1)} \sum_{k=0}^{d_i} \binom{d_i+1}{k+1} (1-q)^{k+1} (q)^{d_i-k} \tag{B.4}$$

$$= \frac{1}{(1-q)(d_i+1)} \sum_{k=1}^{d_i+1} \binom{d_i+1}{k} (1-q)^k (q)^{d_i+1-k} \tag{B.5}$$

$$= \frac{1 - q^{d_i+1}}{(1-q)(d_i+1)} \tag{B.6}$$

Similarly, if the Bernoulli mask corresponding to $j \to i$ is 1, then the total number of incoming messages to node $i$ is given by

$$2 + \sum_{k=1}^{d_i-1} m_k$$

including one self-loop, which is never dropped, as noted earlier. On the other hand, the weight of the edge is simply 0 if the corresponding Bernoulli mask is 0. Using the Law of Total Expectation, the expected weight of the edge $j \to i$ can be computed as follows:

$$\mathbb{E}_{\text{DE}}\left[\hat{\boldsymbol{A}}_{ij}^{(1)}\right] = q \cdot 0 + (1-q) \mathbb{E}_{m_1,\ldots,m_{d_i-1}}\left[\frac{1}{2 + \sum_{k=1}^{d_i-1} m_k}\right] \tag{B.7}$$

$$= (1-q) \sum_{k=0}^{d_i-1} \binom{d_i-1}{k} (1-q)^k (q)^{d_i-1-k} \left(\frac{1}{2+k}\right) \tag{B.8}$$

$$= \sum_{k=0}^{d_i-1} \frac{(d_i-1)!}{(k+2)!(d_i-1-k)!} (1-q)^{k+1} (q)^{d_i-1-k} (k+1) \tag{B.9}$$

$$= \sum_{k=2}^{d_i+1} \frac{(d_i-1)!}{(k)!(d_i+1-k)!} (1-q)^{k-1} (q)^{d_i+1-k} (k-1) \tag{B.10}$$

$$= \frac{1}{d_i(d_i+1)(1-q)} \sum_{k=2}^{d_i+1} \binom{d_i+1}{k} (1-q)^k (q)^{d_i+1-k} (k-1) \tag{B.11}$$

$$= \frac{1}{d_i(d_i+1)(1-q)} \left[(d_i+1)(1-q) - 1 + q^{d_i+1}\right] \tag{B.12}$$

$$= \frac{1}{d_i} \left(1 - \mathbb{E}_{\text{DE}}\left[\hat{\boldsymbol{A}}_{ii}^{(1)}\right]\right) \tag{B.13}$$

$\square$

## B.2 SENSITIVITY IN L-LAYER LINEAR GCNS

**Theorem.** *In an L-layer linear GCN with $\hat{A} = \hat{A}^{asym}$, using DropEdge, DropAgg or DropGNN decreases the sensitivity of a node $i \in \mathcal{V}$ to another node $j \in \mathbb{S}^{(L)}(i)$, thereby reducing its effective receptive field.*

$$\mathbb{E}_{\dots}\left[\left(\hat{A}^L\right)_{ij}\right] = \sum_{(u_0,\dots,u_L)\in\mathsf{Paths}(j\to i)} \prod_{\ell=1}^{L} \mathbb{E}_{\dots}\left[\hat{A}_{u_\ell u_{\ell-1}}\right] < \mathbb{E}_{\mathsf{ND}}\left[\left(\hat{A}^L\right)_{ij}\right] \tag{B.14}$$

*where $\mathsf{ND}$ refers to a NoDrop model ($q = 0$), the placeholder $\cdots$ can be replaced with one of the edge-dropping methods DE, DA or DG, and the corresponding entries of $\mathbb{E}_{\dots}[\hat{A}]$ can be plugged in from Equation 3.3, Equation 3.5 and Equation 3.6, respectively. Moreover, the sensitivity monotonically decreases as the dropping probability is increased.*

*Proof.* Recall that $\dot{P}$ can be viewed as the transition matrix of a non-uniform random walk, such that $\dot{P}_{uv} = \mathbb{P}(u \to v)$. Intuitively, since there is no self-loop on any given L-length path connecting nodes $i$ and $j$ (which are assumed to be L-hops away), the probability of each transition on any path connecting these nodes is reduced. Therefore, so is the total probability of transitioning from $i$ to $j$ in exactly L hops.

More formally, denote the set of paths connecting the two nodes by

$$\mathsf{Paths}(j \to i) = \{(u_0, \dots, u_L) : u_0 = j; u_L = i; (u_{\ell-1} \to u_\ell) \in \mathcal{E}, \forall \ell \in [L]\}$$

The $(i, j)$-entry in the propagation matrix is given by

$$\left(\dot{P}^L\right)_{ij} = \sum_{(u_0,\dots,u_L)\in\mathsf{Paths}(j\to i)} \prod_{\ell=1}^{L} \dot{P}_{u_\ell u_{\ell-1}} \tag{B.15}$$

Since there is no self-loop on any of these paths,

$$\left(\dot{P}^L\right)_{ij} = \sum_{(u_0,\dots,u_L)\in\mathsf{Paths}(j\to i)} \prod_{\ell=1}^{L} \frac{1}{d_{u_\ell}}\left(1 - \frac{1-q^{d_{u_\ell}+1}}{(1-q)(d_{u_\ell}+1)}\right) \tag{B.16}$$

$$< \sum_{(u_0,\dots,u_L)\in\mathsf{Paths}(j\to i)} \prod_{\ell=1}^{L} \left(\frac{1}{d_{u_\ell}+1}\right) \tag{B.17}$$

The right hand side of the inequality is the $(i, j)$-entry in the L$^{\text{th}}$ power of the propagation matrix of a NoDrop model. From Equation 3.5 and Equation 3.6, we know that Equation B.17 is true for DropAgg and DropGNN as well. We conclude the first part of the proof using Equation 3.2 – the sensitivity of node $i$ to node $j$ is proportional to $(\dot{P}^L)_{ij}$.

Next, we recall the geometric series for any $q$:

$$1 + q + \dots + q^d = \frac{1-q^{d+1}}{1-q} \tag{B.18}$$

Each of the terms on the right are increasing in $q$, hence, all the $\dot{P}_{u_\ell u_{\ell-1}}$ factors are decreasing in $q$. Using this result with Equation B.15, we conclude the second part of the theorem.

$\square$

## C TEST-TIME MONTE-CARLO AVERAGING

In Section 3, we focused on the expected sensitivity of the stochastic representations in models using DropEdge-like regularization strategies. This corresponds to their training-time behavior, wherein the activations are random. At test-time, the standard practice is to turn these methods off by setting

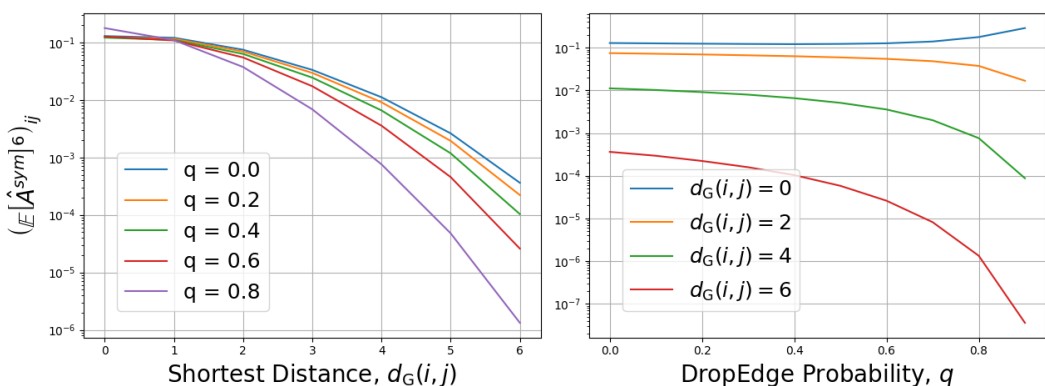

**Figure 4:** Entries of $\ddot{\boldsymbol{P}}^6$, averaged after binning node-pairs by their shortest distance.

$q = 0$. However, this raises the over-smoothing levels back up (Xuanyuan et al., 2023). Another way of making predictions is to perform multiple stochastic forward passes, as during training, and then averaging the model outputs. This is similar to Monte-Carlo Dropout, which is an efficient way of ensemble averaging in MLPs (Gal & Ghahramani, 2016b), CNNs (Gal & Ghahramani, 2016a) and RNNs (Gal & Ghahramani, 2016c). In addition to alleviating over-smoothing, this approach also outperforms the standard implementation (Xuanyuan et al., 2023). We can study the effect of random edge-dropping in this setting by examining the sensitivity of the *expected representations*:

$$\left\| \frac{\partial}{\partial \boldsymbol{x}_j} \mathbb{E}\left[ \boldsymbol{z}_i^{(L)} \right] \right\|_1$$

In linear models, the order of the two operations – expectation and sensitivity computation – is irrelevant:

$$\mathbb{E}\left[ \left\| \frac{\partial \boldsymbol{z}_i^{(L)}}{\partial \boldsymbol{x}_j} \right\|_1 \right] = \left\| \mathbb{E}\left[ \hat{\boldsymbol{A}}_{ij} \right] \boldsymbol{W} \right\|_1 = \left\| \mathbb{E}\left[ \frac{\partial \boldsymbol{z}_i^{(L)}}{\partial \boldsymbol{x}_j} \right] \right\|_1 = \left\| \frac{\partial}{\partial \boldsymbol{x}_j} \mathbb{E}\left[ \boldsymbol{z}_i^{(L)} \right] \right\|_1 \tag{C.1}$$

In general, the two quantities can be related using the convexity of norms and Jensen's inequality:

$$\left\| \frac{\partial}{\partial \boldsymbol{x}_j} \mathbb{E}\left[ \boldsymbol{z}_i^{(L)} \right] \right\|_1 \leq \mathbb{E}\left[ \left\| \frac{\partial \boldsymbol{z}_i^{(L)}}{\partial \boldsymbol{x}_j} \right\|_1 \right] \tag{C.2}$$

Therefore, the discussion in the previous subsections extends to the MC-averaged representations as well. Although tighter bounds may be derived for this setting, we will leave that for future works.

## D    Additional Figures

In this section, we present some additional figures that demonstrate the negative effects of random edge-dropping, particularly focusing on providing empirical evidence for scenarios not covered by the theory in Section 3.

### D.1    Symmetrically Normalized Propagation Matrix

The results in Section 3.1 correspond to the use of $\hat{\boldsymbol{A}} = \hat{\boldsymbol{A}}^{\text{asym}}$ for aggregating messages – in each message passing step, only the in-degree of node $i$ is used to compute the aggregation weights of the incoming messages. In practice, however, it is more common to use the symmetrically normalized propagation matrix, $\hat{\boldsymbol{A}} = \hat{\boldsymbol{A}}^{\text{sym}}$, which ensures that nodes with high degree do not dominate the information flow in the graph (Kipf & Welling, 2017). As in Equation 3.2, we are looking for

$$\mathbb{E}_{\mathbf{M}^{(1)}, \ldots, \mathbf{M}^{(L)}}\left[ \prod_{l=1}^{L} \hat{\boldsymbol{A}}^{(\ell)} \right] = \ddot{\boldsymbol{P}}^L$$

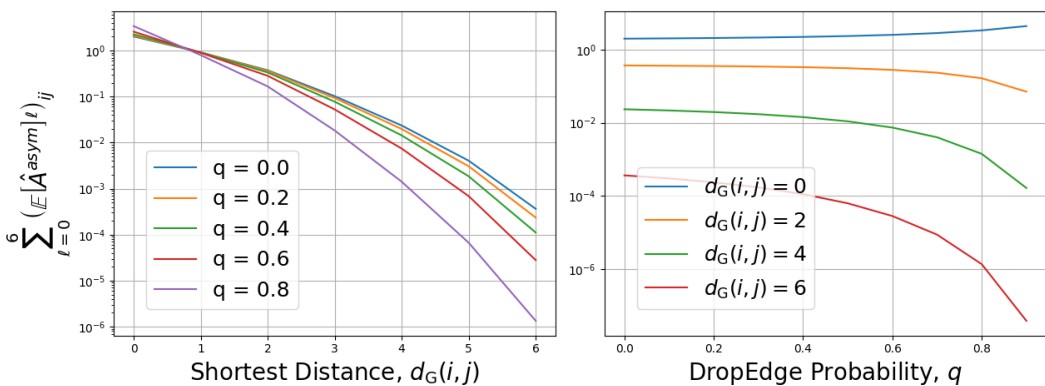

**Figure 5:** Entries of $\sum_{l=0}^{6} \dot{P}^l$, averaged after binning node-pairs by their shortest distance.

where $\ddot{P} := \mathbb{E}_{\text{DE}}[\hat{A}^{\text{sym}}]$. While $\ddot{P}$ is analytically intractable, we can approximate it using Monte-Carlo sampling. Accordingly, we use 20 samples of $\mathbf{M}$ to compute an approximation of $\ddot{P}$, and plot out the entries of $\ddot{P}^L$, as we did for $\dot{P}^L$ in Figure 1. The results are presented in Figure 4, which shows that while the sensitivity between nearby nodes is affected to a lesser extent compared to those observed in Figure 1, that between far-off nodes is significantly reduced, same as earlier.

### D.2    Upper Bound on Expected Sensitivity

Black et al. (2023) showed that the sensitivity between any two nodes in a graph can be bounded using the sum of the powers of the propagation matrix. In Section 3.2, we extended this bound to random edge-dropping methods with independent edge masks smapled in each layer:

$$\mathbb{E}_{\mathbf{M}^{(1)},\ldots,\mathbf{M}^{(L)}}\left[\left\|\frac{\partial z_i^{(L)}}{\partial x_j}\right\|_1\right] = \zeta_3^{(L)}\left(\sum_{\ell=0}^{L} \mathbb{E}\left[\hat{A}\right]^\ell\right)_{ij}$$

Although this bound does not have a closed form, we can use real-world graphs to study its entries. We randomly sample 100 molecular graphs from the Proteins dataset (Dobson & Doig, 2003) and plot the entries of $\sum_{l=0}^{6} \dot{P}^\ell$ (corresponding to DropEdge) against the shortest distance between node-pairs. The results are presented in Figure 5. We observe a polynomial decline in sensitivity as the DropEdge probability increases, suggesting that it is unsuitable for capturing LRIs.

### D.3    Test Accuracy versus DropNode Probability

In Equation 3.4, we noted that the expectation of sensitivity remains unchanged when using DropNode. However, these results were only in expectation. In practice, a high DropNode probability will result in poor communication between distant nodes, preventing the model from learning to effectively model LRIs. This is supported by the results in Table 2, where we observed a negative correlation between the test accuracy and DropNode probability. Moreover, DropNode was the only algorithm which recorded negative correlations on homophilic datasets. In Figure 6, we visualize these relationships, noting the stark contrast with Figure 2, particularly in the trends with homophilic datasets.

## E    Experiment Details

In this section, we expand on the details of the experiment in Section 4.

**Descriptions of the Datasets**. Cora (McCallum et al., 2000) and CiteSeer (Giles et al., 1998) are citation networks – their nodes represent scientific publications and an edge between two nodes indicates that one of them has cited the other. The features of each publication are represented by

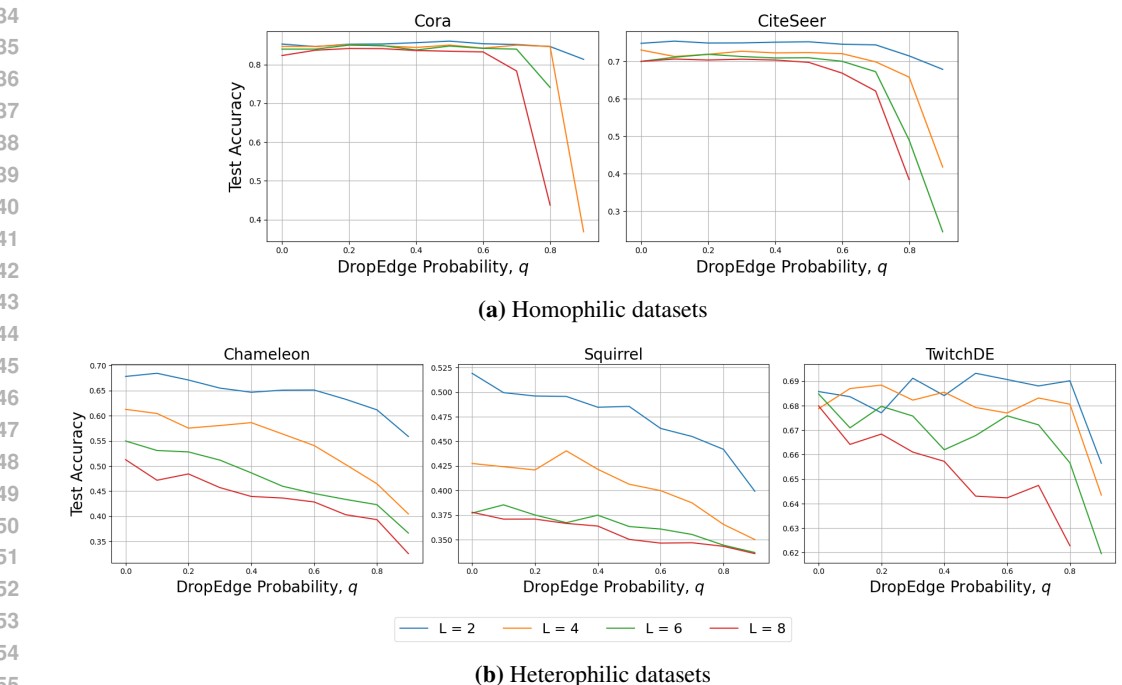

**(a)** Homophilic datasets

**(b)** Heterophilic datasets

**Figure 6:** Dropping probability versus test accuracy of DropNode-GCN.

a binary vector, where each index indicates whether a specific word from a dictionary is present or absent. Several studies have showed that these datasets have high homophily in node labels (Lim et al., 2021; Zhu et al., 2020) and that they are modelled much better by shallower networks than by deeper ones (Zhou et al., 2020). Chameleon and Squirrel (Rozemberczki et al., 2021) are networks of English Wikipedia web pages on the respective topics, and the edges between web pages indicate links between them. The task is to predict the average-monthly traffic on each of the web pages. Finally, TwitchDE (Rozemberczki et al., 2021) is a network of Twitch users in Germany, with the edges between them representing their mutual follower relationships. The node features are embeddings of the games played by the users. The task is to predict whether the users use explicit language.

**Training Hyperparameters**. We standardise most of the hyperparameters across all experiments in order to isolate the effect of dropping probability. Specifically, we fix the size of the hidden representations in each layer at 64, and a linear readout layer is used to compute the node-level logits. The models are trained using the Adam optimizer (Kingma & Ba, 2015), with a learning rate of $3 \times 10^{-3}$ and a weight decay of $5 \times 10^{-4}$, for a total of 300 epochs. For GCN, we use symmetric normalization of the adjacency matrix to compute the edge weights (Kipf & Welling, 2017).

**GAT Runs**. It is quite problematic to train deep GAT models due to vanishing gradients (Dasoulas et al., 2021). Accordingly, we discard the runs where the model fails to learn, and performs just as well as a random classifier. Specifically, we compute the class distribution in each of the networks, and discard the runs where the test performance does not exceed the maximum proportion. This comes out to be 0.3021, 0.2107, 0.2288, 0.2003 and 0.6045 for Cora, CiteSeer, Chameleon, Squirrel and TwitchDE, respectively.

