# OpenReview forum: "Effects of Random Edge-Dropping on Over-Squashing in Graph Neural Networks"
_ICLR.cc/2025/Conference — Submitted to ICLR 2025_

### Official Review · Reviewer_Ku4N · 2024-10-18

**Soundness:** 3
**Presentation:** 3
**Contribution:** 2
**Rating:** 5
**Confidence:** 3

**Summary:**

This paper investigates the effects of DropEdge and other variants on over-squashing problem, by calculating the sensitivity theoretically and emperically.

**Strengths:**

- The motivation is quite clear and solid.
- Overall the writing is pretty good and understandable.
- The paper's logic is sound, and the assumptions are reasonable.
- The empirical study on sensitivity is well-designed and makes great sense.
- The experiments on real world datasets align well with the theory and intuition.

**Weaknesses:**

- The proof in appendix A1 is not quite clear. Maybe explain more.
- There are some datasets on long range interaction (LRI) and over-squashing, say trees match and peptides datasets. They are missing from the experiments.
- The contribution of the paper is marginal. The mainstream of solving over-squashing is by graph rewiring or graph transformers. This paper addresses DropEdge and variants cannot solve over-squashing, that's fine, but empirically that is already true, and no one uses it for LRI.

**Questions:**

See weakness. Can you explain more the derivation in A1?

---

> ### Author Response · Authors · 2024-11-21
>
> Thank you for your review! Kindly find our responses below to the weaknesses and questions raised.
>
> >The proof in appendix A1 is not quite clear. Maybe explain more.
>
> We have added some remarks building up to the computation of the expected message weight in Appendix B.1. Kindly let us know if anything remains unclear.
>
> >There are some datasets on long range interaction (LRI) and over-squashing, say trees match and peptides datasets. They are missing from the experiments.
>
> Reviewer Z8L5 remarked something similar as well. Kindly check our comment addressing these concerns together under the heading “**set of datasets used**”. We assume you are referring to the NeighborsMatch dataset in [1]. We do agree that evaluation on more datasets is always desirable for making generalizable conclusions. In particular, the NeighborsMatch dataset would have been a suitable choice for demonstrating the effect of edge-dropping on model performance in long-range tasks. However, due to the synthetic nature of the dataset, such an experiment would only serve to establish the point that these methods limit a model’s capacity to learn long-range information. Indeed, Figures 1, 4 and 5 establish exactly that, since the model is unable to effectively propagate information over long distances.
>
> As for the peptides datasets (we assume you are referring to the long-range graph benchmark [2]), the problem is that they are graph-level tasks. Specifically, regardless of whether the node representations are learnt as a function of distant nodes or not (i.e., whether over-squashing occurs), the prediction is made by mixing the information from all the nodes, making the effect of over-squashing on model performance hard to disentangle.
>
> We chose the datasets in this work with the aim of highlighting a critical distinction in the results for homophilic and heterophilic datasets, which was suggested by the theory in Section 3. We did not see a similar intent in adding the suggested datasets, which is why we did not include it in our analysis. However, upon your suggestion, we will continue working on this study and add the results for the suggested datasets to the final version of this work; the large size of these datasets prevents us from presenting the results by the end of the rebuttal period.
>
> >The contribution of the paper is marginal. The mainstream of solving over-squashing is by graph rewiring or graph transformers. This paper addresses DropEdge and variants cannot solve over-squashing, that's fine, but empirically that is already true, and no one uses it for LRI.
>
> Reviewer 3v8w remarked something similar as well. Kindly check our comment addressing these concerns together under the heading “**significance of the work**”. We refrain from including the case of graph transformers since it deviates from the common framework shared by the methods we study &ndash; regularizing models with stochastic message passing. We would like to emphasize that our goal was not to challenge the use of methods designed to alleviate over-squashing, but instead those that target over-smoothing, particularly citing their significantly limited evaluation on long-range tasks &ndash; the primary use-case of deep GNNs. Accordingly, we have also not included the graph rewiring methods you mentioned. Although, we do believe that a sensitivity analysis &ndash; similar to the one in our work &ndash; of methods targeting over-squashing can make it clearer whether the performance boost in real-world settings is due to improved long-range information propagation.
>
> While it is true that DropEdge is not typically promoted for modeling LRIs, it is worth noting that no prior work explicitly discourages its use for long-range tasks either. Importantly, related graph rewiring methods aimed at alleviating over-squashing do not benchmark their performance against DropEdge (or variants) on long-range tasks. Despite this, DropEdge remains a widely used regularization method for MPNNs and training deep GNNs. We demonstrate that it is unsuitable for long-range tasks, which are the main use-case practitioners opt deeper GNNs for. This highlights an important gap in current evaluations and emphasizes the need for caution when applying such methods, as their effectiveness heavily depends on task requirements. The literature lacks such negative results, which are essential to assess the broader applicability of these techniques effectively.
>
> [1] Uri Alon and Eran Yahav. On the bottleneck of graph neural networks and its practical implications. In International Conference on Learning Representations, 2021.
>
> [2] Vijay Prakash Dwivedi, Ladislav Rampášek, Mikhail Galkin, Ali Parviz, Guy Wolf, Anh Tuan Luu, and Dominique Beaini. Long range graph benchmark. In Thirty-sixth Conference on Neural Information Processing Systems Datasets and Benchmarks Track, 2022.

---

> > ### Comment · Reviewer_Ku4N · 2024-11-25
> >
> > Thank you for the great effort you put into your work and the rebuttal. I am now convinced of the significance of theoretical understanding dropedge on over-squashing. I have read the comments by other reviewers as well. The other reviewers also agree that the paper needs some improvement in experiments as well as a proposed solution. Therefore, I would like to remain my score.

---

### Official Review · Reviewer_biHW · 2024-10-28

**Soundness:** 1
**Presentation:** 2
**Contribution:** 1
**Rating:** 5
**Confidence:** 4

**Summary:**

The paper investigates the effects of random edge-dropping algorithms, such as DropEdge and its variants (DropNode, DropAgg, and DropGNN), on over-squashing in MPNNs. Over-squashing occurs when information from exponentially growing neighborhoods is compressed into finite-sized node representations, limiting the ability to capture long-range interactions.

**Strengths:**

1. This paper is well written and easy to follow.
2. The paper provides a comprehensive analysis of different types of MPNNs.

**Weaknesses:**

1. I do not find the main experiments of the paper convincing. The authors categorize the experimental results in Table 2 into homophilic datasets and heterophilic datasets to verify that the experimental results align with the theoretical results. In fact, I think that the magnitude of the Spearman correlation values in Table 2 depends on the optimal dropping probability for that dataset, rather than the homogeneity of the dataset. In other homophilic datasets, such as PubMed and ogbn-arxiv, the optimal dropping probability is quite small, and I do not think the same conclusions as those in the paper can be reached.
2. The conclusions described in Lemma 3.2 and Theorem 3.1 of the paper lack quantitative descriptions, which makes the conclusions less compelling. Additionally, the paper does not discuss potential solutions; it only analyzes this phenomenon, which limits the contribution of the work.
3. The paper only discusses DropEdge and its variants. In fact, for graph random dropping methods, Dropout [1,2] is the most commonly used, while DropMessage [3,4] is the most advanced. The lack of exploration of such non-edge dropping methods significantly limits the applicability of this work.
4. The experimental section of the paper is insufficient as it only includes a limited set of node classification tasks. Other downstream tasks, such as link prediction and graph classification, are missing experimental results.


[1] Improving neural networks by preventing co-adaptation of feature detectors.

[2] Dropout: a simple way to prevent neural networks from overfitting

[3] Dropmessage: Unifying Random Dropping for Graph Neural Networks

[4] Unifying Graph Contrastive Learning via Graph Message Augmentation

**Questions:**

Please refer to the points I mentioned in the weakness section.

---

> ### Author Response · Authors · 2024-11-21
>
> Thank you for your review! Kindly find our responses below to the weaknesses and questions raised.
>
> >I do not find the main experiments of the paper convincing. The authors categorize the experimental results in Table 2 into homophilic datasets and heterophilic datasets to verify that the experimental results align with the theoretical results. In fact, I think that the magnitude of the Spearman correlation values in Table 2 depends on the optimal dropping probability for that dataset, rather than the homogeneity of the dataset. In other homophilic datasets, such as PubMed and ogbn-arxiv, the optimal dropping probability is quite small, and I do not think the same conclusions as those in the paper can be reached.
>
> Excellent remark! We do agree that the experiment setup is not flawless, and that if the optimal dropping probability, the correlation can be expected to be negative. In Figure 2a, we show that the test accuracy for Cora and CiteSeer monotonically increases with DropEdge probability. As for PubMed and obgn-arxiv, we will work the results for them right after we have the results for Dropout, DropMessage and SkipNode with the current set of datasets!
>
> In the meantime, can we check which method has low optimal dropping probability for PubMed? We checked the DropEdge paper [1], and the optimal probabilities reported in Table 6 are quite high for PubMed &ndash; 0.3 for GCN, 0.7 for ResGCN, 0.5 for JKNet, 0.5 for IncepGCN, and 0.8 for GraphSage. In the GRAND (DropNode) paper [2], the optimal dropping probability reported is 0.5, with 0.6 dropout rate in the input layer and 0.8 in the hidden layers. As for DropAgg [3], the optimal dropping probabilities are not reported, and for DropGNN [4], they are only reported for synthetic datasets. Kindly let us know what we might have missed.
>
> Regardless, the results reported in these papers usually correspond to the best performing models over multiple training runs. This is in contrast to our experimental design, where we use the average performance over multiple runs to compute the correlation statistics, effectively disregarding the variance in model performance under different dropping methods.
>
> >The conclusions described in Lemma 3.2 and Theorem 3.1 of the paper lack quantitative descriptions, which makes the conclusions less compelling.
>
> Taking your feedback into account, we now direct the reader to the appendix for the quantitative results corresponding to Theorem 3.1. We have presented the quantitative results for Theorem 3.1 in the appendix due to limited space available in the main text. The exact quantitative statement would depend on the choice of dropping method, so we simply summarize the conclusion in the main text.
>
> As for Lemma 3.2, it follows directly from the discussion after Lemma 3.1. Instead of restating the quantitative results, we chose to summarize the conclusion for brevity. The precise statement would depend on the choice of the algorithm used, so the reader may kindly refer to the preceding discussion for further details.
>
> [1] Yu Rong, Wenbing Huang, Tingyang Xu, and Junzhou Huang. Dropedge: Towards deep graph convolutional networks on node classification. In International Conference on Learning Representations, 2020.
>
> [2] Wenzheng Feng, Jie Zhang, Yuxiao Dong, Yu Han, Huanbo Luan, Qian Xu, Qiang Yang, Evgeny Kharlamov, and Jie Tang. Graph random neural networks for semi-supervised learning on graphs. In H. Larochelle, M. Ranzato, R. Hadsell, M.F. Balcan, and H. Lin (eds.), Advances in Neural Information Processing Systems, volume 33, pp. 22092–22103. Curran Associates, Inc., 2020.
>
> [3] Bo Jiang, Yong Chen, Beibei Wang, Haiyun Xu, and Bin Luo. Dropagg: Robust graph neural networks via drop aggregation. Neural Networks, 163:65–74, 2023.
>
> [4] Pál András Papp, Karolis Martinkus, Lukas Faber, and Roger Wattenhofer. Dropgnn: Random dropouts increase the expressiveness of graph neural networks. In M. Ranzato, A. Beygelzimer, Y. Dauphin, P.S. Liang, and J. Wortman Vaughan (eds.), Advances in Neural Information Processing Systems, volume 34, pp. 21997–22009. Curran Associates, Inc., 2021.

---

> ### Author Response · Authors · 2024-11-21
>
> >Additionally, the paper does not discuss potential solutions; it only analyzes this phenomenon, which limits the contribution of the work.
>
> Reviewer Z8L5 remarked something similar as well. Kindly check our comment addressing these concerns together under the heading “**solution for the highlighted problem**”, where we also describe how our theoretical analysis can be used to design a DropEdge variant with node-wise dropping probability set to minimize information loss during message-passing. While we do agree that evaluating such a method could strengthen our work, several methods have already been introduced to address this problem (see Appendix A.2 and A.3). Instead, our goal was to instead highlight the problem of limited evidence supporting the suitability of methods designed for alleviating over-smoothing in tackling long-range tasks &ndash; the primary use-case of deep GNNs.
>
> >The paper only discusses DropEdge and its variants. In fact, for graph random dropping methods, Dropout is the most commonly used, while DropMessage is the most advanced. The lack of exploration of such non-edge dropping methods significantly limits the applicability of this work.
>
> Reviewer Z8L5 remarked something similar as well. Kindly check our comment addressing these concerns together under the heading “**set of methods evaluated**”. In particular, while we will not be able to include a theoretical analysis of Dropout, DropMessage and SkipNode by the end of the rebuttal period, we are working on adding the empirical results for them. We hope that would address your concern.
>
> >The experimental section of the paper is insufficient as it only includes a limited set of node classification tasks. Other downstream tasks, such as link prediction and graph classification, are missing experimental results.
>
> In graph-level tasks, regardless of whether the node representations are learnt as a function of distant nodes or not (i.e., whether over-squashing occurs), the prediction is made by mixing the information from all the nodes, making the effect of over-squashing on model performance hard to disentangle. In other words, even if information fails to mix over distant nodes via message-passing, it may get mixed by the pooling layer.
>
> The effect of a method on over-squashing can be expected to be closely related to link prediction tasks. However, we have not come across any long-range link prediction datasets; kindly let us know if we missed out on any.
>
> We wish to emphasize that our goal is not to simply compare these methods across a range of datasets, but rather verify their suitability towards long-range tasks in particular. The effect of a method on over-squashing should be closely related to the model performance on long-range node-classification tasks, but that conclusion is hard to extend to other problem types, eg. graph-level tasks.

---

> ### Comment · Reviewer_biHW · 2024-11-23
>
> Thank you for the author's patient responses. I have thoroughly read all the author's replies as well as the feedback from other reviewers. While the author has provided responses to most of the issues, many of them still require substantial experimental validation. Although the textual content shows potential directions for improvement, these have not been fully implemented yet. Therefore, I believe this paper still requires significant revisions. Additionally, in my opinion, a specialized version of DropMessage is likely the most feasible solution, and the author may consider delving deeper into this technique.

---

### Official Review · Reviewer_Z8L5 · 2024-10-29

**Soundness:** 2
**Presentation:** 3
**Contribution:** 2
**Rating:** 5
**Confidence:** 4

**Summary:**

This paper investigates the impact of random edge-dropping techniques on the issue of over-squashing in GNNs, specifically in the context of message-passing neural networks (MPNNs). While previous studies have explored edge-dropping methods to address over-smoothing, this paper highlights that these methods can exacerbate over-squashing, thus hindering GNNs’ performance on long-range tasks. Through theoretical analysis and empirical validation, the authors demonstrate that edge-dropping methods reduce the sensitivity of distant nodes to each other, leading to limited message-passing across long distances. The results suggest that while random edge-dropping methods like DropEdge improve performance in short-range tasks, they deteriorate performance in long-range tasks due to overfitting to local structures.

**Strengths:**

- This work provides a fresh perspective by focusing on over-squashing, a less-explored limitation of edge-dropping techniques in GNNs

- The paper presents a comprehensive theoretical framework, supported by experiments that confirm the detrimental impact of edge-dropping on long-range node interactions.

- The paper provides clear explanations showing how random edge-dropping improves accuracy in short-range tasks but worsens performance in long-range tasks.

**Weaknesses:**

- While the paper identifies limitations of random edge-dropping methods—which are not specifically designed to address over-squashing—it lacks a proposed solution or guidance on mitigating over-squashing for these methods.

- The choice of datasets could be improved. It may be beneficial for the authors to follow the experimental setup outlined in [1].

- Some relevant works are omitted: (1) SkipNode [2], which selectively skips nodes at each layer, and (2) DropMessage [3], which introduces a generalizable framework for dropout-based techniques in GNNs.

- The source codes and the used dasets should be provided to facilitate the reproducibility of this work. Please refer to https://anonymous.4open.science.


Refs:

[1] On the bottleneck of graph neural networks and its practical implications.

[2] SkipNode: On alleviating performance degradation for deep graph convolutional networks

[3] Dropmessage: Unifying random dropping for graph neural networks

**Questions:**

Please see the weaknesses listed above.

---

> ### Author Response · Authors · 2024-11-21
>
> Thank you for your review! Kindly find our responses below to the weaknesses and questions raised.
>
> >While the paper identifies limitations of random edge-dropping methods—which are not specifically designed to address over-squashing—it lacks a proposed solution or guidance on mitigating over-squashing for these methods.
>
> Reviewer biHW remarked something similar as well. Kindly check our comment addressing these concerns together under the heading “**solution for the highlighted problem**”, where we also describe how our theoretical analysis can be used to design a DropEdge variant with node-wise dropping probability set to minimize information loss during message-passing. In particular, while we do agree that proposing such a method would strengthen our work, several methods have already been introduced to address this problem (see Appendix A.2 and A.3). Instead, our goal was to highlight the problem of limited evidence supporting the suitability of methods designed for alleviating over-smoothing in tackling long-range tasks &ndash; the primary use-case of deep GNNs.
>
> >The choice of datasets could be improved. It may be beneficial for the authors to follow the experimental setup outlined in [1].
>
> Reviewer Ku4N remarked something similar as well. Kindly check our comment addressing these concerns together under the heading “**set of datasets used**”. In particular, while we do appreciate the value of extensive evaluation on a variety of datasets, the size of the datasets used in [1], in addition to the short rebuttal period, we will unfortunately be unable to evaluate on the suggested datasets. However, we will continue working on this study and add the results for the suggested datasets to the final version of this work.
>
> >Some relevant works are omitted: (1) SkipNode [2], which selectively skips nodes at each layer, and (2) DropMessage [3], which introduces a generalizable framework for dropout-based techniques in GNNs.
>
> Reviewer biHW remarked something similar as well. Kindly check our comment addressing these concerns together under the heading “**set of methods evaluated**”. In particular, while we will not be able to include a theoretical analysis of Dropout, DropMessage and SkipNode by the end of the rebuttal period, we are working on adding the empirical results for them. We hope that would address your concern.
>
> >The source codes and the used datasets should be provided to facilitate the reproducibility of this work. Please refer to https://anonymous.4open.science.
>
> While we do acknowledge that the linked repository-anonymization tool is quite popular, we are wary of using it because of the extensive set of write permissions it requests to all public and private repository data. This includes the code, settings, webhooks and services, deploy keys and collaboration invites. We don’t see why such extensive access is necessary. If you have any suggestion for safely sharing the code, kindly let us know.
>
> Please note that we will share the code as soon as the final scores are released, regardless of the outcome since this submission will be de-anonymized at that point anyway.

---

### Official Review · Reviewer_3v8w · 2024-10-31

**Soundness:** 3
**Presentation:** 3
**Contribution:** 2
**Rating:** 6
**Confidence:** 4

**Summary:**

The paper studies the effect that randomly dropping edges (in different ways) has on over-squashing and more broadly on the propagation of information. The conclusion is that techniques like DropEdge have a negative effect on the sensitivity between distant nodes, but possibly a positive effect on closer nodes. This is shown both mathematically and through experiments.

**Strengths:**

The paper is well presented and well written. There is a good amount of literature review and context provided as well. The authors study different scenarios and overall provide a pretty detailed mathematical understanding of the topic they wish to study.

**Weaknesses:**

I believe that the main weakness is that the core claim of the paper is not surprising. It is clear in general that removing edges will decrease the sensitivity. In fact, the converse is a motivation for graph rewiring techniques — adding the smallest amount of edges to maximise the propagation of information.

From another point of view, I believe that this trade-off is clear also when contrasting over-squashing and over-smoothing. A large spectral gap means better sensitivity, but also means more smoothing. For this reason,  removing edges (which will likely lower the spectral gap) will have the effect of combatting over-smoothing, by reducing the connectivity (meaning global sensitivity is lowered).

I do believe that the work is well-written and quite thorough. For this reason I do not wish to be overly negative in my initial scoring, although I would need to be convinced during the rebuttal phase on the strength of the contribution.

It is also worth pointing out that there already are a number of works [e.g. 1, 2] that link the trade offs between over-smoothing to over-squashing.

[1] Revisiting Over-smoothing and Over-squashing Using Ollivier-Ricci Curvature. ICML 2023
[2] On the Trade-off between Over-smoothing and Over-squashing in Deep Graph Neural Networks. CIKM 2023

**Questions:**

Related to the weakness, I would appreciate if the authors could comment on why they believe the main claim to be a surprise to readers working in related areas?

Could the authors comment on how the findings are different from works such as [1, 2] that already comment on the trade-off between over-squashing and over-smoothing, especially [2] connecting it to spectral quantities (see my comment in the weaknesses section above on  spectral gaps).

[1] Revisiting Over-smoothing and Over-squashing Using Ollivier-Ricci Curvature. ICML 2023
[2] On the Trade-off between Over-smoothing and Over-squashing in Deep Graph Neural Networks. CIKM 2023

---

> ### Author Response · Authors · 2024-11-21
>
> Thank you for your review! Kindly find our responses below to the weaknesses and questions raised.
>
> >I believe that the main weakness is that the core claim of the paper is not surprising. It is clear in general that removing edges will decrease the sensitivity.
>
> Reviewer Ku4N remarked something similar as well. Kindly check our comment addressing these concerns together under the heading “**significance of the work**”. Notably, we found that edge-dropping can increase sensitivity between neighboring nodes, as shown in Figures 1, which is an unintuitive result in general. This empirical observation helps explain why deep GNNs perform well on short-range tasks with DropEdge, DropNode, DropGNN and DropAgg, despite the model-dataset misalignment &ndash; the reduced receptive fields guide the model to prioritize short-range signals, and prevent overfitting to the long-range ones.
>
> We show that conclusions on the efficacy of edge-dropping methods in regularizing models may not be trivially extended to long-range tasks. The literature lacks such negative results which emphasize the need for practicing caution when employing these methods when training deep GNNs. Our hope is to draw attention to the evaluation of other methods designed for alleviating over-smoothing and training deep GNNs (see Appendix A.1) being limited to short-range tasks, where such deep models are arguably unnecessary. It is important to critically reassess these methods with a renewed focus on long-range tasks &ndash; the more relevant use case of deep GNNs.
>
> >In fact, the converse is a motivation for graph rewiring techniques — adding the smallest amount of edges to maximize the propagation of information.
>
> We do agree that addition of edges has been shown to be necessary for alleviating over-squashing. The motivation for our work was to challenge the widespread adoption of Dropout and DropEdge for regularizing GNNs. We intended to establish that methods designed for alleviating over-smoothing have not been evaluated in a systematic manner, particularly on long-range tasks (understandably, because over-squashing was discovered more recently). Therefore, practitioners must exercise caution when adopting these methods for training deep GNNs. In the case of edge-dropping methods, this means ensuring that the ground-truth labels do not rely on long-range information propagation.
>
> >Could the authors comment on how the findings are different from works such as [1, 2] that already comment on the trade-off between over-squashing and over-smoothing, especially [2] connecting it to spectral quantities.
>
> The focus of our work is not to provide new insights into the over-smoothing vs. over-squashing trade-off itself. In fact, we drew inspiration from these works and sought to address a critical, overlooked question: *Given the known trade-off between over-smoothing and over-squashing, might methods designed to alleviate over-smoothing be unsuited for long-range tasks &ndash; the primary motivation for using deep GNNs?* To that end, we investigate how dropping probability and node degrees impact over-squashing, and how such methods affect model performance in long-range tasks. While [1, 2] do predict that edge-dropping methods would exacerbate the over-squashing problem, we precisely characterize the dependence of this phenomenon on the dropping probability and the degree sequence of the network, and demonstrate the negative effects on performance in common real-world tasks.
>
> [1] Khang Nguyen, Hieu Nong, Vinh Nguyen, Nhat Ho, Stanley Osher, and Tan Nguyen. Revisiting over-smoothing and over-squashing using ollivier-ricci curvature. In Proceedings of the 40th International Conference on Machine Learning, ICML’23. JMLR.org, 2023.
>
> [2] Jhony H. Giraldo, Konstantinos Skianis, Thierry Bouwmans, and Fragkiskos D. Malliaros. On the trade-off between over-smoothing and over-squashing in deep graph neural networks. In Proceedings of the 32nd ACM International Conference on Information and Knowledge Management, CIKM ’23, pp. 566–576, New York, NY, USA, 2023. Association for Computing Machinery.

---

> > ### Comment · Reviewer_3v8w · 2024-11-23
> > **Thanks!**
> >
> > Thank you for the clarifications. I think that the work is interesting and well-written. I believe that the key ideas presented do contribute to a better and more nuanced understanding of phenomena related to over-squashing and their relationship to edge-droppping. I do however believe that the work is quite specialised as it primarily focuses on edge-dropping techniques, so I would have appreciated it if it could have also been a bit more general than this.
> >
> > I would be grateful if the authors could comment on if they believe the works provides a better understanding of GNNs more generally than when it comes to edge-dropping. For instance, it seems like this could be connected to techniques that involve the sampling of neighborhoods when the neighborhoods become too large (i.e. GraphSAGE).

---

> > > ### Author Response · Authors · 2024-11-25
> > >
> > > Thank you for the feedback. Kindly find our responses below:
> > >
> > > >I do however believe that the work is quite specialised as it primarily focuses on edge-dropping techniques, so I would have appreciated it if it could have also been a bit more general than this.
> > >
> > > We do recognize that edge-dropping algorithms form a small subset of available methods for training deep GNNs. Accordingly, we have added results for Dropout, DropMessage and SkipNode, as some other reviewers' suggested. We hope that addresses your concern, and provides additional evidence for the prevalence of the problem we are trying to bring to light.
> > >
> > > >I would be grateful if the authors could comment on if they believe the works provides a better understanding of GNNs more generally than when it comes to edge-dropping. For instance, it seems like this could be connected to techniques that involve the sampling of neighborhoods when the neighborhoods become too large (i.e. GraphSAGE).
> > >
> > > GraphSAGE's neighborhood sampling can be seen as a variant of DropEdge -- while the former samples a fixed number of neighbors in each message-passing step, the latter samples the neighbors iid. Therefore, we do believe that our conclusions would also extend to GraphSAGE, as you mentioned. Not only that, even residual connections (ResGCN) may be unsuited for long-range tasks, not by reducing the strength of signal over long distances, but rather by amplifying the strength of short-range signals (see our recent comment on additional results).

---

> > > > ### Comment · Reviewer_3v8w · 2024-11-26
> > > >
> > > > Thank you for your response, I am happy to increase my score if some coverage of methods like GraphSAGE is added (even in the Appendix with a pointer in the main text). I believe that this could be seen as a way to make the work much more general and would add significant value in my opinion. I am happy even if this is left as a high-level discussion.

---

> > > > > ### Author Response · Authors · 2024-11-29
> > > > > **Thank you for the suggestion!**
> > > > >
> > > > > Although we were unable to update the manuscript in time, we will add the following section to the appendix of the final version, with a link in the main text pointing to it.
> > > > >
> > > > > **Extension to Methods Employing Architectural Modification**
> > > > >
> > > > > Our exploration in this work has been mainly limited to stochastic regularization methods, including edge-dropping methods, and related ones like Dropout, DropMessage and SkipNode. We may follow a similar intuition to study other methods like as well. For example, GraphSAGE operates by sampling a subset of neighbors for message aggregation in each step. From an over-squashing perspective, it is very similar to DropEdge and therefore, we can expect a similar decline in sensitivity between distant nodes when using GraphSAGE.
> > > > >
> > > > > On the other hand, residual connection reduce the *influence* of distant nodes by increasing the influence of nearby neighbors. For example, in the case of ResGCN, the operation $\mathbf{Z}^{(\ell+1)} = \hat{\mathbf{A}}\mathbf{Z}^{(\ell)}\mathbf{W}^{(\ell)} + \mathbf{Z}^{(\ell)}$ first collects information from nodes $\ell$-hops away, like in a regular GCN model, and then adds the information from nodes up to $(\ell-1)$-hops away, effectively decreasing the influence of the nodes $\ell$-hops away. In other words, we may expect residual connections to be similarly unsuited for long-range tasks, not by reducing the strength of the signal over long distances, but rather by amplifying the strength of short-range signals. In [Figure X](https://drive.google.com/file/d/1B8i2HB-5Zkq4TbHJ8mdT_dC1g5VpY2LW/view?usp=sharing), we present the empirical influence distribution [Y] of a 6-layer ResGCN, computed for the Cora network. Here, we can clearly see that residual connections significantly reduce the influence of distant nodes on final representations, and can thus be expected to be unsuitable for long-range tasks.
> > > > >
> > > > > [Y] Keyulu Xu, Chengtao Li, Yonglong Tian, Tomohiro Sonobe, Ken-ichi Kawarabayashi, and Stefanie Jegelka. Representation learning on graphs with jumping knowledge networks. In Proceedings of the 35th International Conference on Machine Learning, volume 80 of Proceedings of Machine Learning Research, pp. 5453–5462. PMLR, 07 2018.

---

> > > > > > ### Comment · Reviewer_3v8w · 2024-12-02
> > > > > >
> > > > > > Thanks, this sounds good. I have increased my score from 5 to 6.
> > > > > >
> > > > > > I do think that the work has fundamentally interesting ideas and the paper seems polished and easy to follow. It contributes to a deeper understanding of over-squashing.
> > > > > >
> > > > > > I believe the paper would still benefit from a broader positioning outside of just edge dropping, which the authors have now added to some extent.

---

### Author Response · Authors · 2024-11-21
**Common Response to the Reviewers' Feedback**

We sincerely appreciate the feedback from the reviewers. We will use this response to address some of the shared concerns.

**Significance of the work**. Reviewers 3v8w and Ku4N mention that it is not surprising for random edge-dropping methods to exacerbate the over-squashing problem, and that they are not promoted for long-range tasks. While we do agree that it might be intuitive for such methods to reduce sensitivity between distant nodes, our work theoretically characterizes the precise dependence of this decay on dropping probability and the degree distribution of the graph. We believe that such a precise characterization can guide the design of dropping methods that keep over-squashing in check. For example, one may design a DropEdge variant with node-wise dropping probability set to ensure that the message weight over each edge does not decrease by more than a preset threshold, say 0.9. We leave such an exploration for future works since extensive research is already aimed at addressing over-squashing using graph rewiring, and we wish to highlight a different problem. The aim of our work was *not* to simply present results for edge-dropping methods, but rather to draw attention to the limited evaluation protocol employed for a suite of algorithms designed for training deep GNNs. The focus has largely been limited to short-range tasks like citation networks (eg. Cora, CiteSeer, PubMed), recommendation networks (eg. ogbn-products) and social networks (eg. karate, Reddit). However, the primary use-case of deep GNNs is in long-range tasks, and trainability on short-range tasks is of little significance.

A trade-off between over-smoothing and over-squashing has been established in the literature, especially when using graph rewiring techniques (see Appendix A.3). Our work draws inspiration from these results and sought to address a critical, overlooked question: *Given the known trade-off between over-smoothing and over-squashing, might methods designed to alleviate over-smoothing be unsuited for long-range tasks?* We demonstrate the prevalence of this problem through poor test-time performance of methods designed for training deep GNNs in common tasks involving heterophilic networks. This highlights an important gap in current evaluations and emphasizes the need for caution when applying such methods, as their effectiveness heavily depends on task requirements. The literature lacks such negative results, which are essential to assess the broader applicability of these techniques effectively.

Our hope is to draw attention to the importance of critically assessing similar methods, such as normalization techniques and residual connections (see Appendix A.1). Understanding their effects on over-squashing and long-range tasks &ndash; the more relevant use case of deep GNNs &ndash; is essential for assessing their broader applicability. Some other methods designed for alleviating over-smoothing, eg. NodeNorm [1] and PairNorm [2], might not come at the cost of higher over-squashing levels. For long-range tasks, practitioners must seek such techniques instead of popular choices like Dropout or DropEdge.

Finally, our analysis is also expected to inspire a study on other graph rewiring methods, including those introduced for alleviating over-squashing (see Appendix A.2 and A.3), eg. CurvDrop [3] and BORF [4]. While theoretically motivated, a sensitivity analysis of these methods remains largely lacking, which makes it unclear whether the performance boost in real-world settings is indeed due to improved long-range information propagation, as intended, or perhaps something else.

[1] Kuangqi Zhou, Yanfei Dong, Kaixin Wang, Wee Sun Lee, Bryan Hooi, Huan Xu, and Jiashi Feng. Understanding and resolving performance degradation in deep graph convolutional networks. In Proceedings of the 30th ACM International Conference on Information & Knowledge Management, pp. 2728&ndash;2737, 2021b.

[2] Lingxiao Zhao and Leman Akoglu. Pairnorm: Tackling oversmoothing in gnns. In International Conference on Learning Representations, 2020.

[3] Yang Liu, Chuan Zhou, Shirui Pan, Jia Wu, Zhao Li, Hongyang Chen, and Peng Zhang. Curvdrop: A ricci curvature based approach to prevent graph neural networks from over-smoothing and over- squashing. In Proceedings of the ACM Web Conference 2023, WWW ’23, pp. 221&ndash;230, New York, NY, USA, 2023. Association for Computing Machinery.

[4] Jake Topping, Francesco Di Giovanni, Benjamin Paul Chamberlain, Xiaowen Dong, and Michael M. Bronstein. Understanding over-squashing and bottlenecks on graphs via curvature. In International Conference on Learning Representations, 2022.

---

> ### Author Response · Authors · 2024-11-21
> **Common Response to the Reviewers' Feedback (Continuation)**
>
> **Solution for the highlighted problem**. Reviewers Z8L5 and biHW raise concerns about the lack of a proposed solution. In Section 3, we precisely characterize the dependence of sensitivity on the dropping probability and the degree sequence of the graph. Future works may exploit these relations to design new methods that systematically drop edges while limiting over-squashing levels. For example, one may design a DropEdge variant with node-wise dropping probability set to ensure that the message weight over each edge does not decrease by more than a preset threshold, say 0.9. We leave such an exploration for future works since extensive research is already aimed at addressing over-squashing using graph rewiring, and we wish to highlight a different problem.
>
> While we do agree that proposing such a method would strengthen our work, several methods have already been introduced to address this problem (see Appendix A.2 and A.3). Instead, our goal was to highlight the problem of limited evidence supporting the suitability of methods designed for alleviating over-smoothing in tackling long-range tasks &ndash; the primary use-case of deep GNNs. The closest example to the methods we analyze is CurvDrop, which addresses the over-squashing problem by selectively dropping edges based on their curvatures. This prevalence of the problem we highlight can be seen in the evaluation of CurvDrop, which is limited to citation networks, a synthetic SBM dataset and the karate network &ndash; all short-range tasks.
>
> **Set of methods evaluated**. Reviewers Z8L5 and biHW suggested adding methods like Dropout, DropMessage and SkipNode, which aim to tackle the problem of over-smoothing as well. We did recognize the relevance of these methods, but refrained from including them in order to maintain a focus on edge-dropping methods, which none of these methods are a type of. While we will not be able to include a theoretical analysis of these methods by the end of the rebuttal period, we are working on adding the empirical results for them.
>
> **Set of datasets used**. Reviewers Z8L5 and Ku4N raise a concern about the limited set of datasets used for evaluation. We do agree that evaluation on more datasets is always desirable for making generalizable conclusions. We chose the datasets in this work with the aim of highlighting a critical distinction in the results for homophilic and heterophilic datasets, which was suggested by the theory in Section 3. We do see the value in testing the edge-dropping methods with the suggested large scale datasets, however since we perform 200 runs (10 dropping probabilities, 4 model depths, 5 runs each) for each model-datasets-dropout combination, we will be unable to present the results for these datasets by the end of the rebuttal period. However, we will continue working on this study and add the results for the suggested datasets to the final version of this work.

---

### Author Response · Authors · 2024-11-25
**Additional Results**

**Updates to the Manuscript**

1. We mistakenly characterized the decay rate of sensitivity as super-exponential, when it is actually polynomial. We have fixed this in the manuscript.

2. We have added some additional explanation for the proof of Lemma 3.2 upon the suggestion of reviewer Ku4N.

3. We have added a related works section in the appendix. Appendix A.1 discusses the over-smoothing methods that we believe can benefit from further evaluation, especially with an emphasis on long-range tasks. Appendix A.2 discusses methods designed for alleviating over-squashing, and Appendix A.3 discusses the trade-off between over-smoothing and over-squashing, as well as the methods designed to tackle the two problem simultaneously.

**Additional Results** (not in the manuscript)

1. We evaluated Dropout, DropMessage and SkipNode, following a similar protocol as in Section 4. The results presented below corroborate our findings about stochastic dropping methods:
| Method | GNN | Cora | CiteSeer | Chameleon | TwitchDE | Squirrel |
| ------ | --- | ---- | -------- | -------- | -------- | -------- |
| Dropout | GCN | +0.034 | -0.309 | -0.610 | -0.465 | -0.300 |
| Dropout | GAT | +0.120 | -0.110 | -0.836 | -0.545 | -0.397 |
| DropMessage | GCN | -0.220 | -0.607 | -0.404 | -0.364 | -0.199 |
| DropMessage | GAT | +0.028 | -0.368 | -0.530 | -0.697 | -0.148 |
| SkipNode | GCN | +0.161 | +0.017 | -0.528 | -0.285 | -0.244 |
| SkipNode | GAT | +0.403 | +0.466 | -0.436 | +0.248 | +0.278 |

2. We also expect residual connections (ResGCN) to be similarly unsuited for long-range tasks, not by reducing the strength of signal over long distances, but rather by amplifying the strength of short-range signals (cf. influence distribution [1]). We present the empirical influence distribution of a 6-layer ResGCN, computed for the Cora network, [here](https://drive.google.com/file/d/1B8i2HB-5Zkq4TbHJ8mdT_dC1g5VpY2LW/view?usp=sharing) (shared using a throw-away account). Here, we can clearly see that residual connections significantly reduce the influence of distant nodes on final representations, and can thus be expected to be unsuitable for long-range tasks.

[1] Keyulu Xu, Chengtao Li, Yonglong Tian, Tomohiro Sonobe, Ken-ichi Kawarabayashi, and Stefanie Jegelka. Representation learning on graphs with jumping knowledge networks. In Proceedings of the 35th International Conference on Machine Learning, volume 80 of Proceedings of Machine Learning Research, pp. 5453–5462. PMLR, 07 2018.

---

### Meta-Review · Area_Chair_poHR · 2024-12-21

**Metareview:**

**(a) Scientific Claims and Findings:**
The paper investigates the impact of random edge-dropping techniques, such as DropEdge, on the over-squashing phenomenon in Graph Neural Networks (GNNs). While these methods have been effective in mitigating over-smoothing and facilitating the training of deep GNNs, their influence on over-squashing has remained unexplored. The authors present theoretical analyses demonstrating that random edge-dropping exacerbates over-squashing, thereby degrading model performance on tasks requiring long-range dependencies. Empirical evaluations on real-world datasets corroborate these findings, showing that although DropEdge variants enhance performance in short-range tasks, they deteriorate performance in long-range tasks. The study concludes that random edge-dropping reduces the effective receptive field of GNNs, which, while beneficial for short-range tasks, misaligns models for long-range interactions, leading to overfitting to short-range artifacts in the training data and poor generalization.

**(b) Strengths:**
* Theoretical Insight: The paper provides a theoretical framework that elucidates the negative impact of random edge-dropping on over-squashing in GNNs.
* Empirical Evaluation: The authors conduct extensive experiments on five real-world (relatively small-scale) datasets, offering empirical evidence that supports their theoretical claims and highlighting the practical implications of their findings.
* Critical Re-evaluation of Existing Methods: The study prompts a reassessment of widely adopted techniques like DropEdge and its variants, especially concerning their suitability for tasks involving long-range dependencies.

**(c) Weaknesses:**
* Novelty of insight: It is well known and intuitive that dropping edges increases over-squashing (as also noted by Reviewers 3v8w and Ku4N). The authors argue that they provide a precise characterisation of the effect of randomly dropping edges. However, they analyse the effect on a proxy measure that has been related to over-squashing (i.e. sensitivity) and not the generalization performance of GNNs. Furthermore, the quantification is limited to linear GCNs.
* Lack of Proposed Solutions (Minor): While the paper identifies and analyzes the problem of exacerbated over-squashing due to random edge-dropping, it does not propose alternative methods or solutions to address this issue, leaving the reader without guidance on how to mitigate the identified problem. Y
* Limited Scope of Analysis: The focus is primarily on random edge-dropping techniques; the paper does not explore other potential factors contributing to over-squashing or consider a broader range of methods that might alleviate this issue. Even though the main story evolves on over-squashing and long range tasks, the studied datasets are mostly small scale and not related to this issue.
* Limited Scope of Theory: The theory is limited to linear GCNs and average effects on sensitivity. It does not really quantify the scale of the problem.
* Absence of Practical Recommendations: The study highlights the detrimental effects of certain techniques but falls short of providing actionable recommendations for practitioners seeking to balance the trade-offs between over-smoothing and over-squashing in GNNs.

**(d) Reasons for Rejection:**
After careful consideration, the decision to reject the paper is based on the following reasons:
1. Narrow Focus Without Broader Context: While the analysis of random edge-dropping is interesting and thorough, the paper does not situate its findings within a broader context of GNN training challenges, potentially limiting its relevance to a wider audience. The theoretical insight is relatively well known and, considering this fact, the theoretical quantification of the random edge dropping effect not sufficiently realistic, to create a high enough impact. The experiments cannot compensate for the limitation of the theory, as they do quantify the detrimental effect on a wide range of tasks, disentangling the potentially positive effect of over-smoothing  from the detrimental effect of over-squashing.
Addressing these concerns by proposing mitigation strategies, offering practical guidance, or situating the findings within a broader context would enhance the paper's contribution and its potential for acceptance in future submissions.
Alternatively, the authors have posed an interesting question during the rebuttal: “Given the known trade-off between over-smoothing and over-squashing, might methods designed to alleviate over-smoothing be unsuited for long-range tasks – the primary motivation for using deep GNNs?”
It’s more general analysis looking at a wide range of different methods supporting their hypothesis could also be of merit, but it would require a more comprehensive study  of different approaches.

**Additional Comments On Reviewer Discussion:**

While all reviewers appreciate the thorough rebuttal, three out of four reviewers remain unconvinced that the paper should be accepted at this stage. For instance, Reviewer biHW has requested a substantial additional experimental validation and suggested that a specialized version of DropMessage could be the most feasible direction to offer a solution. Also Reviewer Reviewer Ku4N would appreciate a proposal of a solution to overcome the detrimental effect on long-range tasks. Reviewer 3v8w would like to see a wider analysis of different methods, Reviewer Z8L5 and Reviewer Ku4N suggest to broaden the consideration of more and different datasets (more related to over-squashing).

---

### Decision · Program_Chairs · 2025-01-22

Reject